# Analysis of the impact of coronavirus disease 19 on hospitalization rates for chronic non-communicable diseases in Brazil

**Rafael Alves Guimarães**[1,2]*, **Gabriela Moreira Policena**[2], **Hellen da Silva Cintra de Paula**[3], **Charlise Fortunato Pedroso**[3], **Raquel Silva Pinheiro**[2], **Alexander Itria**[4], **Olavo de Oliveira Braga Neto**[5], **Adriana Melo Teixeira**[5], **Irisleia Aires Silva**[5], **Geraldo Andrade de Oliveira**[6], **Karla de Aleluia Batista**[3,7]

**1** Faculty of Nursing, Federal University of Goiás, Goiânia, Goiás, Brazil, **2** Institute of Tropical Pathology and Public Health, Federal University of Goiás, Goiânia, Goiás, Brazil, **3** Federal Institute of Education, Science and Technology of Goiás, Goiânia Oeste Campus, Goiânia, Goiás, Brazil, **4** Federal University of São Carlos, Sorocaba Campus, Sorocaba, São Paulo, Brazil, **5** Department of Hospital and Emergency Care of the Ministry of Health, Distrito Federal, Brasília, Brazil, **6** Federal Institute of Education, Science and Technology of Goiás, Valparaíso Campus, Valparaíso, Goiás, Brazil, **7** Institute of Biological Sciences 2, Federal University of Goiás, Goiânia, Goiás, Brazil

* rafaelalves@ufg.br

## Abstract

### Background

The coronavirus disease (COVID-19) pandemic has impacted health services and healthcare systems worldwide. Studies have shown that hospital admissions for causes related to chronic non-communicable diseases (NCDs) have decreased significantly during peak pandemic periods. An analysis of the impact of the COVID-19 pandemic on hospital admissions for NCDs is essential to implement disability and mortality mitigation strategies for these groups. Therefore, this study aimed to analyze the impact of the COVID-19 pandemic on hospital admissions for NCDs in Brazil according to the type of NCD, sex, age group, and region of Brazil.

### Methods

This is an ecological study conducted in Brazil. Data on hospital admissions from January 1, 2017 to May 31, 2021 were extracted from the Unified Health System's Hospital Admissions Information System. The hospital admission rates per 100,000 thousand inhabitants were calculated monthly according to the type of NCD, sex, age group, and region of Brazil. Poisson regression models were used to analyze the impact of the COVID-19 pandemic on the number of hospital admissions. In this study, the pre-pandemic period was set from January 1, 2017 to February 29, 2020 and the during-pandemic from March 1, 2020 to May 31, 2021.

### Results

There was a 27.0% (95.0%CI: -29.0; -25.0%) decrease in hospital admissions for NCDs after the onset of the pandemic compared to that during the pre-pandemic period.

**Data Availability Statement:** All files are available from the SIH-SUS database (http://tabnet.datasus.gov.br/cgi/tabcgi.exe?sih/cnv/niuf.def).

**Funding:** Brazilian Health Ministry, process grant number 25000.038957/2020-10. The funders had no role in study design, data collection and analysis, decision to publish, or preparation of the manuscript.

**Competing interests:** The authors have declared that no competing interests exist.

Decreases were found for all types of NCDs—cancer (-23.0%; 95.0%CI: -26.0; -21.0%), diabetes mellitus (-24.0%; 95.0%CI: -25.0%; -22.0%), cardiovascular diseases (-30.0%; 95.0% CI: -31.0%; -28.0%), and chronic respiratory diseases (-29.0%; 95.0%CI: -30.0%; -27.0%). In addition, there was a decrease in the number of admissions, regardless of the age group, sex, and region of Brazil. The Northern and Southern regions demonstrated the largest decrease in the percentage of hospital admissions during the pandemic period.

## Conclusions

There was a decrease in the hospitalization rate for NCDs in Brazil during the COVID-19 pandemic in a scenario of social distancing measures and overload of health services.

## Introduction

Brazil is the third leading country heavily impacted by the coronavirus disease (COVID-19) pandemic with respect to the number of cases, only after the United States and India. In Brazil, COVID-19 has caused high morbimortality and high costs for society in general that are leading to unprecedented social and economic impacts, aggravating socioeconomic disparities. In country, the first case and death due to COVID-19 were reported on February 26 and March 17, 2020, respectively [1]. Since then, 25,620,209 cases and 628,067 deaths due to COVID-19 have been confirmed in Brazil [1]. The high volume of COVID-19 cases across the country can be primarily attributed to the low incidence of testing, lack of adherence to isolation and social distancing measures at the federal level, and delay in the national COVID-19 vaccination campaign [2]. Moreover, novel variants of severe acute respiratory syndrome coronavirus 2 (SARS-CoV-2), such as P.1. (gamma) emerging from the Brazilian Amazon and Omicron of the South Africa, have contributed to the disease's high transmissibility in the country [3, 4].

Over the course of the pandemic, most countries have adopted strategies to limit virus mobility (e.g., lockdowns) and have suspended non-essential economic activities, resulting in school and university closures. Large-scale events have been banned, and restrictions have been implemented to limit the movement of people at specific times to reduce the transmission of SARS-CoV-2 and the consequential strain on healthcare services [5–8]. Studies have shown that lockdown and social distancing measures are effective strategies for reducing the prevalence of COVID-19 and the rates of hospitalizations and mortality due to COVID-19, especially in conjunction with other preventive measures [5, 7, 9].

In Brazil, a national lockdown was not instituted by the federal government. However, measures to restrict the movement of people and suspend non-essential activities were gradually and distinctly enacted by governors and mayors of different Brazilian states and the Federal District [5, 8]. The Federal District was the first region in the country to promote social distancing immediately after the onset of the COVID-19 pandemic was declared on March 11, 2020 by suspending massive events and educational activities [5, 8]. Some municipalities implemented the blocking measures before the pandemic was declared in early March to contain the increase in cases [5, 8]. Subsequently, all Brazilian states implemented social distancing measures until the end of March 2020, including suspension of in-person events and classes, quarantine for the groups most vulnerable to serious COVID-19 outcomes, a full or partial economic standstill, transportation restrictions, or quarantine of the population [5, 8]. Most Brazilian states adopted these strategies for a period of 1–10 days after notification of the first COVID-19 case. Some areas, mainly states in the Northern and Northeastern regions, adopted these measures early

on, even before notification of the first local case. Seven states suspended all economic activities during the first 13 days after notification of the first case in these areas [5].

The literature has thoroughly evidenced that individuals with chronic non-communicable diseases (NCDs), such as cancer, diabetes mellitus, cardiovascular diseases (CVDs), and chronic respiratory diseases (CRDs), are at high risk for negative outcomes related to COVID-19, including need for hospitalization and ventilatory support and death [10, 11]. Studies have shown that the excess number of deaths among individuals with NCDs during the pandemic cannot be solely attributed to the lethality of COVID-19; adoption of social distancing strategies has also contributed to mortality in this population. Social distancing strategies have affected patient management in health services, resulting in a lower demand for access to health services, reduced number of appointments, and suspension of care activities, elective primary care, and highly complex health procedures. Additionally, overwhelming of health services by patients suspected of having COVID-19 have compromised the care and treatment of patients with NCDs, especially in low- and middle-income countries [12–17].

Studies conducted in developed and developing countries have reported a decrease in hospital admissions for causes related to NCDs, such as cancer, diabetes mellitus, CVD, and CRD during the COVID-19 pandemic [13, 16–20]. The Brazilian healthcare system, mainly the public system (Unified Health System [SUS]), was substantially affected by the COVID-19 pandemic [20]. This event occurred heterogeneously across the states and regions of the country. As observed in other countries, there was a downward trend in NCD-related hospital admissions in Brazil during the pandemic [16, 17, 20], which explains the decline in access to health services and less-than-ideal care for people with NCD-related health problems. In patients with NCDs, limited access to health services can result in increased rates of complications, disabilities, and a higher mortality, aggravating the syndemic between COVID-19 and NCDs [21–23].

Brazil is still one of the major centers of the COVID-19 pandemic, although cases have fallen significantly after vaccinations began in January 2021. Although studies have indicated that COVID-19 has resulted in reduced hospital admissions for NCDs, few national investigations have analyzed the effect of this reduction, especially according to type of the NCD, sex, age group, and region of Brazil. Additionally, few studies have compared the number of admissions during the period prior to the pandemic with that during the pandemic using models adjusted for important covariates, such as sex, age group, and potential seasonality in hospital admissions for NCDs. Thus, this study was designed to help better understand the impact of the COVID-19 pandemic on hospital admissions for NCDs in Brazil by verifying the potential effect of social distancing strategies on these individuals' access to health care. These findings may support the implementation of public policies and decision making for better management of NCDs during pandemics. This study aimed to analyze the impact of the COVID-19 pandemic on hospital admissions for NCDs in Brazil, with additional analysis according to the type of NCD, sex, age group, and region of the country to determine whether the COVID-19 pandemic had different impacts on these subgroups.

## Materials and methods

### Study design

This is an ecological of the impact of the COVID-19 pandemic on NCD-related hospital admissions in the Brazilian public health system (SUS).

### Setting

The study was conducted in the Federal District and in the 26 states of the Brazilian federation, which are grouped into five major regions—North (seven states), Northeast (nine states), Midwest (three states and the Federal District), Southeast (three states), and South (three states).

## Population

The study population included adults aged ≥20 years who had been hospitalized for NCD-related causes in the SUS from January 1, 2017 to May 31, 2021. NCDs were classified according to the International Statistical Classification of Diseases and Related Health Problems, Tenth Revision (CID-10), using the following codes: cancer (C00–C97), diabetes mellitus (E10–E14), CVD (I00–I99), and CRD (J30–J98). NCDs were grouped according to the recommendations of the World Health Organization and previous studies [24, 25].

The unit of analysis was the number of hospital admissions for NCD-related causes in adults, which were aggregated by month and year, sex, age group, and region of Brazil. Cases of hospital admissions with unknown data on sex and age group were excluded.

## Data source

Data was retrieved from the Hospital Admissions Information System (SIH) of the SUS on August 16, 2021. This information system was used to process the production relative to hospital admissions in the SUS, and the Hospital Admission Authorization was the source document. The SIH data contain records of all admissions funded by the SUS, thus allowing us to analyze the overall utilization of health services and impact of interventions on morbimortality indicators for all causes. The SIH-SUS is one of the largest national health information systems, recording data from approximately 11.5 million hospitalizations per year [26]. We also used population data obtained from the Brazilian Institute of Geography and Statistics to extract information on resident populations [27] and project the population according to sex, age group, and region of Brazil using arithmetic interpolation.

## Variables

The following indicators were calculated: (i) hospital admission rate for all types of NCDs, (ii) hospital admission rate related to cancer, (iii) hospital admission rate related to diabetes mellitus, (iv) hospital admission rate related to CVD, and (v) hospital admission rate related to CRD. These rates were estimated monthly. Admission rates for each condition were defined as the number of hospital admissions for the respective condition divided by the resident population. To calculate the number of cases per 100,000 residents, these rates were multiplied by 100,000.

The monthly number of hospital admissions for each type of NCD was the dependent variable. The following were the independent variables: (i) sex (male or female), age group (20–39 years, 30–39 years, 40–49 years, 50–59 years, 60–69 years, 70–79 years, or ≥80 years), (iii) region of Brazil (North, Northeast, South, Southeast, or Midwest), and a "dummy" variable of "0" for the pre-pandemic period (January 1, 2017 to February 29, 2020) and "1" for the pandemic period (March 1, 2020 to May 31, 2021). Although the first case in Brazil was reported on February 26, 2020, because of a small number of confirmed cases during the last 4 days of February 2020, we considered the onset of the pandemic period as March 1, 2020. The month of hospital admission was also used as an independent variable to control for possible seasonal variations in hospital admissions over time.

## Data analysis

We used STATA, version 16.0, to analyze the data. Hospital admissions are presented as absolute numbers and rates for the total period. Rates were also provided for each month and stratified by the type of NCD, sex, age group, and region of Brazil. Furthermore, we have presented the absolute number, average number of cases, and average rate of hospital admissions per

100,000 inhabitants during the pre-pandemic (January 1, 2017 to February 29, 2020) and pandemic (March 1, 2020 to May 31, 2021) periods.

Initially, we compared the pre-pandemic and pandemic periods using Student's t-test for independent samples. Next, Poisson regression models with robust variance were applied to analyze the impact of the COVID-19 pandemic on hospitalizations for NCDs. The dependent variable included in the models was the monthly number of hospitalizations for NCDs. Each model was adjusted for sex, age group, region of Brazil, the dummy variable representing the impact of the COVID-19 pandemic ("0," pre-pandemic period; "1," pandemic period), and the month of hospitalization to control for possible seasonal variations in outcomes. The monthly population was used as the exposure variable. We also created stratified models to verify the impact of the pandemic on subgroups according to sex, age group, and region of Brazil. The results of the models are presented as incidence rate ratio (IRR), 95.0% confidence interval (95.0% CI), and regression coefficient. P-values <0.05 were considered statistically significant.

In our study, we used a Poisson regression model with robust variance as used in other studies that evaluated the impact of the COVID-19 pandemic on hospital admissions [28–31]. As mentioned, this model allowed including gender, age group, region of Brazil, the dummy variable representing the impact of the COVID-19 pandemic ("0", pre-pandemic period; "1", pandemic period) and the month of admission to control for possible seasonal variations as dependent variables. Studies show that the Poisson regression model is more suitable for time series count data when compared to the classical interrupted time series (ITS) model [32]. In addition, comparative analysis showed similar results between this model and the ITS in the analysis of interventions [18]. This model was also used instead of the negative binomial model since the data for all outcomes are not over dispersed. All assumptions of the Poisson model were met: (i) dependent variable—count per unit of time and/or space; (ii) observations independent of each other; (iii) the distribution of counts follows a Poisson distribution; (iii) model mean and variance are similar, without overdispersion. Overdispersion was analyzed for all dependent variables by Person Chi-Square for dispersion (p-values of the models ranged from 0.99 to 1.04, indicating the absence of overdispersion) [33].

### Ethical considerations

The data used in this study are available in anonymized secondary and public databases, preventing identification of participants and other sensitive variables. Thus, the requirement for ethical approval and informed consent was waived for this study, in compliance with Resolution 510/16 of the National Health Council of the Ministry of Health.

## Results

From January 1, 2017 to May 31, 2021, there were 9,504,213 hospital admissions for NCD-related causes in Brazil—4,650,367 (48.9%) were related to CVD, 3,261,659 (34.3%) were related to cancer, 1,068,633 (11.2%) were related to CRD, and 523,554 (5.5%) were related to diabetes mellitus. The monthly average number of admissions for NCD-related causes for the total period was 179,324.8. The means for each specific NCD were as follows: 87,742.8 for CVD, 61,540.7 for cancer, 20,162.9 for CRD, and 9,878.4 for diabetes mellitus.

There was a decrease in the mean hospitalization rate for NCDs after the onset of the pandemic in March 2020 (Fig 1 and Table 1). There was a 27.8% decrease in the average hospital admission rate for all NCDs, which decreased from 261.8 (pre-pandemic period) to 189.1 cases per 100,000 inhabitants (pandemic period). In addition, the mean hospital admission rates for all the types of NCD decreased-cancer (72.0 vs. 55.6 cases per 100,000 inhabitants; 22.0% decrease), diabetes mellitus (14.0 vs. 10.0 cases per 100,000 inhabitants; 28.6% decrease),

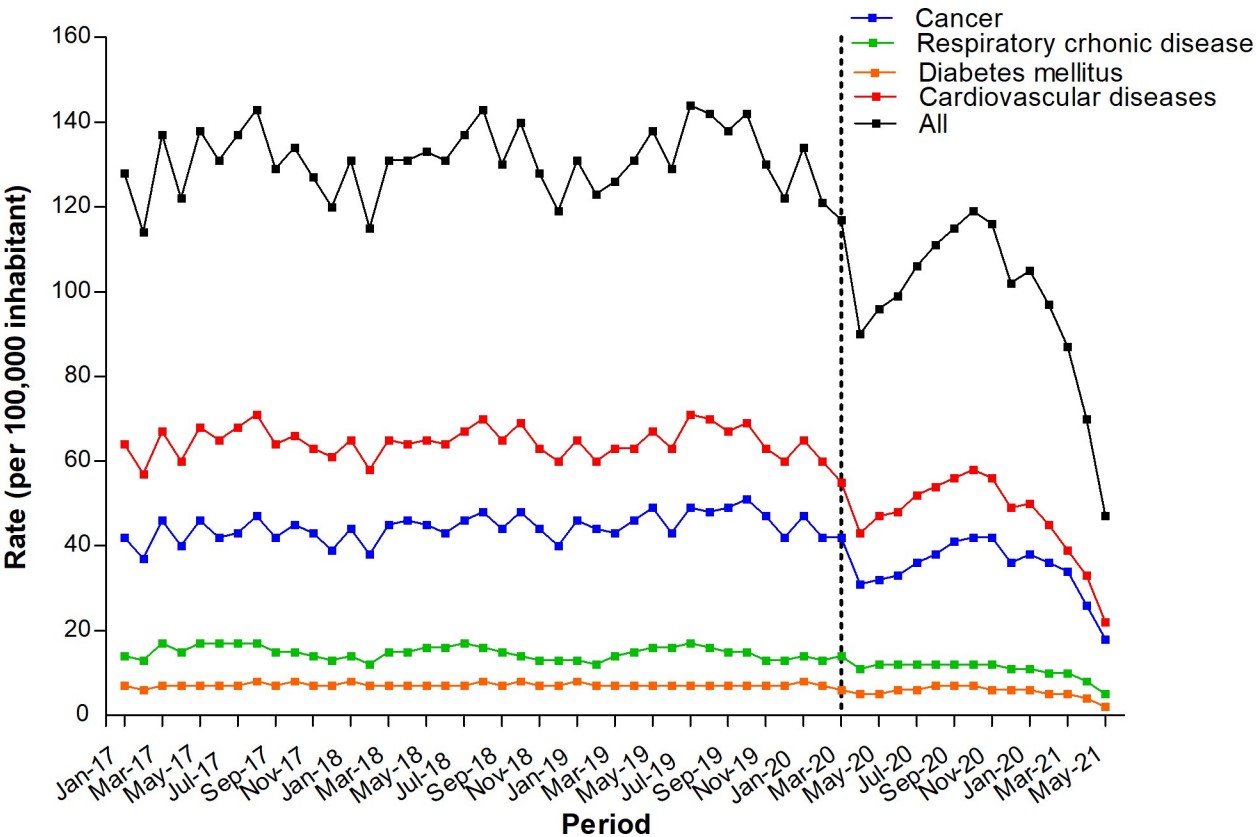

**Fig 1. Hospital admission rate for chronic non-communicable diseases (NCDs) in Brazil according to the type of NCD.** Pre-pandemic period: January 1, 2017 to February 29, 2020; Pandemic period: March 1, 2020 to May 31, 2021.

CVD (141.6 vs. 100.0 cases per 100,000 inhabitants; 29.4% decrease), and CRD (34.2 vs. 23.2 cases per 100,000 inhabitants; 32.2% decrease) (Table 1).

Furthermore, stratified analysis revealed a decrease in the mean hospital admission rate for NCDs during the pandemic period, regardless of sex (Fig 2 and Table 2). There was a 26.0% decrease in the mean hospital admission rate for male patients for all NCDs, which decreased from 291.0 (pre-pandemic period) to 215.3 cases per 100,000 inhabitants (pandemic period). There was a 30.0% decrease in the mean hospital admission rate for female patients for all NCDs (pre-pandemic period vs. pandemic period: 235.8 vs. 162.8 cases per 100,000 inhabitants). We also verified a significant decline in hospital admission rates for cancer, diabetes mellitus, CVD, and CRD, regardless of sex, during the pandemic period (Table 2).

There was a decrease in the average hospital admission rate for all types of NCD during the pandemic period from the pre-pandemic period, regardless of age group (Fig 3 and Table 3)— 20–39 years (from 31.7 to 22.9 cases per 100,000 inhabitants; 27.8% decrease), 40–49 years (from 90.6 to 64.2 cases per 100,000 inhabitants; 29.0% decrease), 50–59 years (from 167.1 to 125.5 cases per 100,000 inhabitants; 24.9% decrease), 60–69 years (from 308.7 to 230.0 cases per 100,000 inhabitants; 25.4% decrease), 70–79 years (from 478.3 to 342.9 cases per 100,000 inhabitants; 28.2 decrease), and ≥80 years (from 724.4 to 515.0 cases per 100,000 inhabitants; 28.7% decrease) (Table 3).

In addition, there was a decrease in the average hospital admission rate for all types of NCD during the pandemic period from the pre-pandemic period, regardless of region of Brazil (Fig 4 and Table 4)—North (from 233.2 to 149.1 cases per 100,000 inhabitants; 36.1% decrease),

**Table 1. Hospital admissions for chronic non-communicable diseases (NCDs) in Brazil before and after the onset of the coronavirus disease pandemic according to the type of NCD.**

| Variables | Pre-pandemic | | Pandemic | | Δ - % (95.0%CI) | p-value* |
|---|---|---|---|---|---|---|
| | Mean | 95.0% CI | Mean | 95.0% CI | | |
| **Cancer** | | | | | | |
| All cases, sum | 2,471,199 | | 790,460 | | | |
| Cases | 4,645 | 4,407–4,883 | 3,764 | 3,445–4,083 | -19.0(-21.8;-16.4) | <0.001 |
| Rate (per 100,000) | 72.0 | 67.4–76.6 | 55.6 | 49.6–61.5 | -22.0(-26.4;-19.7) | <0.001 |
| **Diabetes mellitus** | | | | | | |
| All cases, sum | 396,927 | | 126,627 | | | |
| Cases | 746 | 711–781 | 603 | 554–652 | -19.2(-22.1;-16.5) | <0.001 |
| Rate (per 100,000) | 14.0 | 12.9–15.0 | 10.0 | 8.9–11.4 | -28.6(-31.0;-24.0) | <0.001 |
| **Cardiovascular diseases** | | | | | | |
| All cases, sum | 3,587,156 | | 1,063,211 | | | |
| Cases | 6,742 | 6,414–7,070 | 5,062 | 4,615–5,510 | -24.9(-28.0;-22.1) | <0.001 |
| Rate (per 100,000) | 141.6 | 129.0–154.2 | 100.0 | 85.3–115.1 | -29.4(-33.9;-25.4) | <0.001 |
| **Chronic respiratory disease** | | | | | | |
| All cases, sum | 823,239 | | 245,394 | | | |
| Cases | 1547 | 1,498–1,596 | 1,168 | 1,103–1,233 | -24.5(-26.4;-22.7) | <0.001 |
| Rate (per 100,000) | 34.2 | 30.7–37.6 | 23.2 | 19.5–26.9 | -32.2(-36.5;-28.5) | <0.001 |
| **All types** | | | | | | |
| All cases, sum | 7,278,521 | | 2,225,692 | | | |
| Cases | 13,681 | 13,096–14,266 | 10.598 | 9,785–11,411 | -22.5(-25.3;-20.0) | <0.001 |
| Rate (per 100,000) | 261.8 | 240.6–283.0 | 189.1 | 164.9–214.2 | -27.8(-31.5;-24.3) | <0.001 |

Note: The mean number and rate of hospital admissions were compared between the pre-pandemic (January 1, 2017 to February 29, 2020) and pandemic (March 1, 2020 to May 31, 2021) periods.

95.0% CI = 95.0% confidence interval; *Student's t-test for independent samples.

Midwest (from 259.6 to 118.7 cases per 100,000 inhabitants; 27.1% decrease), Northeast (from 254.34 to 178.0 cases per 100,000 inhabitants; 30.0% decrease), South (from 360.3 to 251.7

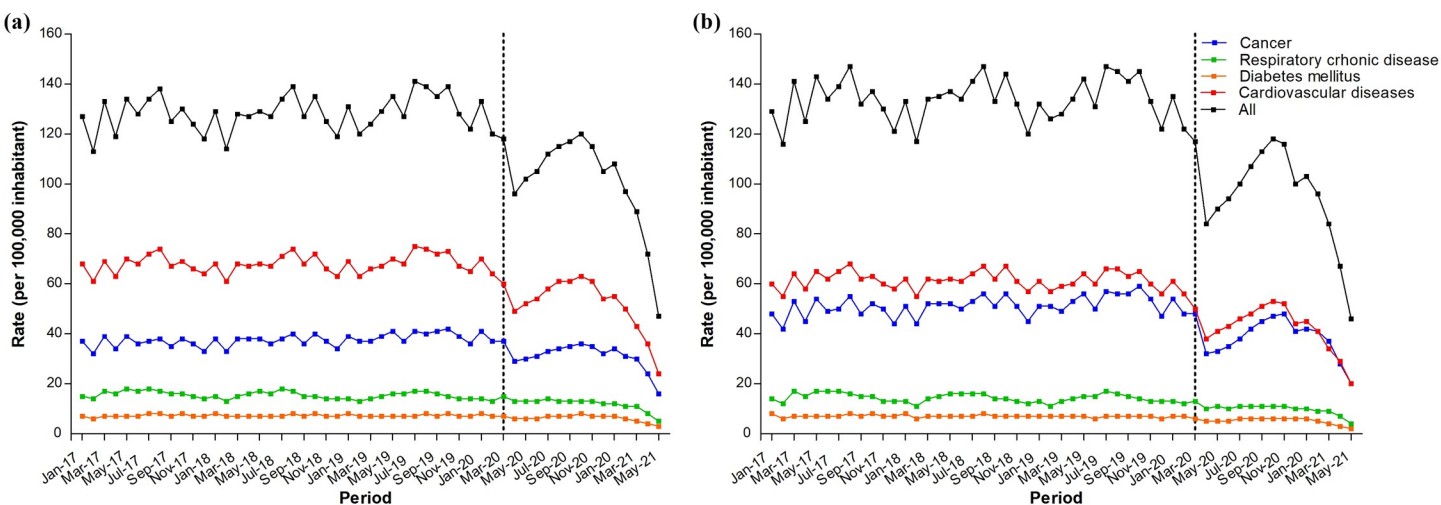

**Fig 2.** Hospital admission rate for chronic non-communicable diseases (NCDs) in Brazil according to the type of NCD and sex: (a) male and (b) female. Pre-pandemic period: January 1, 2017 to February 29, 2020; Pandemic period: March 1, 2020 to May 31, 2021.

**Table 2. Hospital admissions for chronic non-communicable diseases (NCDs) in Brazil before and after the onset of the coronavirus disease pandemic according to the type of NCD and sex.**

| Variables | Male | | | | | | Female | | | | | |
|---|---|---|---|---|---|---|---|---|---|---|---|---|
| | Pre-pandemic | | Pandemic | | Δ - % (95.0% CI) | p-value* | | Pre-pandemic | | Pandemic | | Δ - % (95.0% CI) | p-value* |
| | Mean | 95.0% CI | Mean | 95.0% CI | | | Mean | 95.0% CI | Mean | 95.0% CI | | |
| **Cancer** | | | | | | | | | | | | |
| All cases, sum | 1,014,437 | | 342,700 | | | | 1,456,762 | | 447,760 | | | |
| Cases | 3,814 | 3,519–4,108 | 3,263 | 2,834–3,692 | -14.4 (-19.5; -10.3) | 0.046 | 5,476 | 5,129–5,823 | 4,264 | 3,806–4,722 | -22.1(-25.8; -18.9) | <0.001 |
| Rate (per 100,000) | 76.7 | 68.5–85.0 | 60.5 | 49.9–71.0 | -21.1 (-27.2; -16.5) | 0.029 | 67.3 | 62.2–71.4 | 50.7 | 45.1–56.2 | -24.7(-27.5; -21.3) | <0.001 |
| **Diabetes mellitus** | | | | | | | | | | | | |
| All cases, sum | 197,751 | | 67,607 | | | | 199,176 | | 59,020 | | | |
| Cases | 743 | 689–797 | 644 | 561–726 | -13.3 (-18.7; -8.9) | 0.051 | 748 | 703–793 | 562 | 506–617 | -24.9(-28.0; -22.2) | <0.001 |
| Rate (per 100,000) | 14.8 | 13.2–16.4 | 11.4 | 9.5–13.4 | -23.0 (-28.0; -18.3) | 0.018 | 13.1 | 11.7–14.6 | 8.8 | 7.2–10.3 | -32.8(-38.5; -29.4) | <0.001 |
| **Cardiovascular diseases** | | | | | | | | | | | | |
| All cases, sum | 1,838,911 | | 572,657 | | | | 1,748,245 | | 490,554 | | | |
| Cases | 6,913 | 6,383–7,443 | 5,454 | 4,723–6,184 | -21.1 (-26.0; -16.9) | 0.003 | 6,472 | 6,183–6,961 | 4,671 | 4,153–5,190 | -27.8(-32.8; -25.5) | <0.001 |
| Rate (per 100,000) | 160.2 | 140.3–180.0 | 116.0 | 92.4–139.6 | -27.6(-34.1; -22.5) | 0.012 | 123.0 | 107.9–138.2 | 84.4 | 66.4–102.3 | -31.8(-38.5; -25.9) | 0.005 |
| **Chronic respiratory diseases** | | | | | | | | | | | | |
| All cases, sum | 416,916 | | 131,032 | | | | 406,323 | | 114,362 | | | |
| Cases | 1,567 | 1,491–1,639 | 1,247 | 1,149–1,346 | -20.4(-22.9; -17.9) | <0.001 | 1,527 | 1,461–1,593 | 1,089 | 1,003–1,174 | -28.7(-31.4; -26.3) | <0.001 |
| Rate (per 100,000) | 39.4 | 33.7–45.1 | 27.3 | 21.1–33.5 | -30.7(-37.4; -25.7) | <0.001 | 29.0 | 25.1–32.8 | 19.1 | 15.0–23.1 | -34.1(-40.2; -29.6) | 0.003 |
| **All types** | | | | | | | | | | | | |
| All cases, sum | 3,468,015 | | 1,113,996 | | | | 3,810,506 | | 1,111,696 | | | |
| Cases | 13,037 | 12,102–13,972 | 10,609 | 9,285–11,933 | -18.6(-23.3; -14.6) | 0.005 | 14,325 | 13,625–15,024 | 10,587 | 9,619–11,555 | -26.1(-29.4; -23.1) | <0.001 |
| Rate (per 100,000) | 291.0 | 256.2–326.0 | 215.3 | 173.5–257.0 | -26.0(-32.3; -21.2) | 0.016 | 232.5 | 208.6–256.3 | 162.8 | 134.7–190.2 | -30.0(-35.4; -25.8) | <0.001 |

Note: The mean number and rate of hospital admissions were compared between the pre-pandemic (January 1, 2017 to February 29, 2020) and pandemic (March 1, 2020 to May 31, 2021) periods. 95.0% CI = 95.0% confidence interval; *Student's t-test for independent samples.

cases per 100,000 inhabitants; 30.1% decrease), and Southeast (from 253.6 to 177.6 cases per 100,000 inhabitants; 24.6% decrease) (Table 4).

Table 5 presents results of the Poisson multiple regression models adjusted for sex, age group, region of Brazil, month, and the dummy variable indicating the impact of the pandemic. Compared to the pre-pandemic period, there were significant decreases in admissions related to cancer (-23.0%; 95.0%CI: -26.0; -21.0%), diabetes mellitus (-24.0%; 95.0%CI: -25.0; -22.0%), cardiovascular diseases (-30.0%; 95.0%CI: -31.0%; -28.0%), and chronic respiratory diseases (-29.0%; 95.0%CI: -30.0%; -27.0%). Considering all types of NCDs, there was a 27.0% decline (95.0% CI = -29.0%; -25.0%) in NCD-related hospital admissions after the onset of the COVID-19 pandemic. There was a greater decrease in CRD-related hospital admissions than in cancer-related (29.0% vs. 23.0%) and diabetes mellitus-related (29.0% vs. 24.0%) hospital admissions during the pandemic period from the pre-pandemic period. There was an even

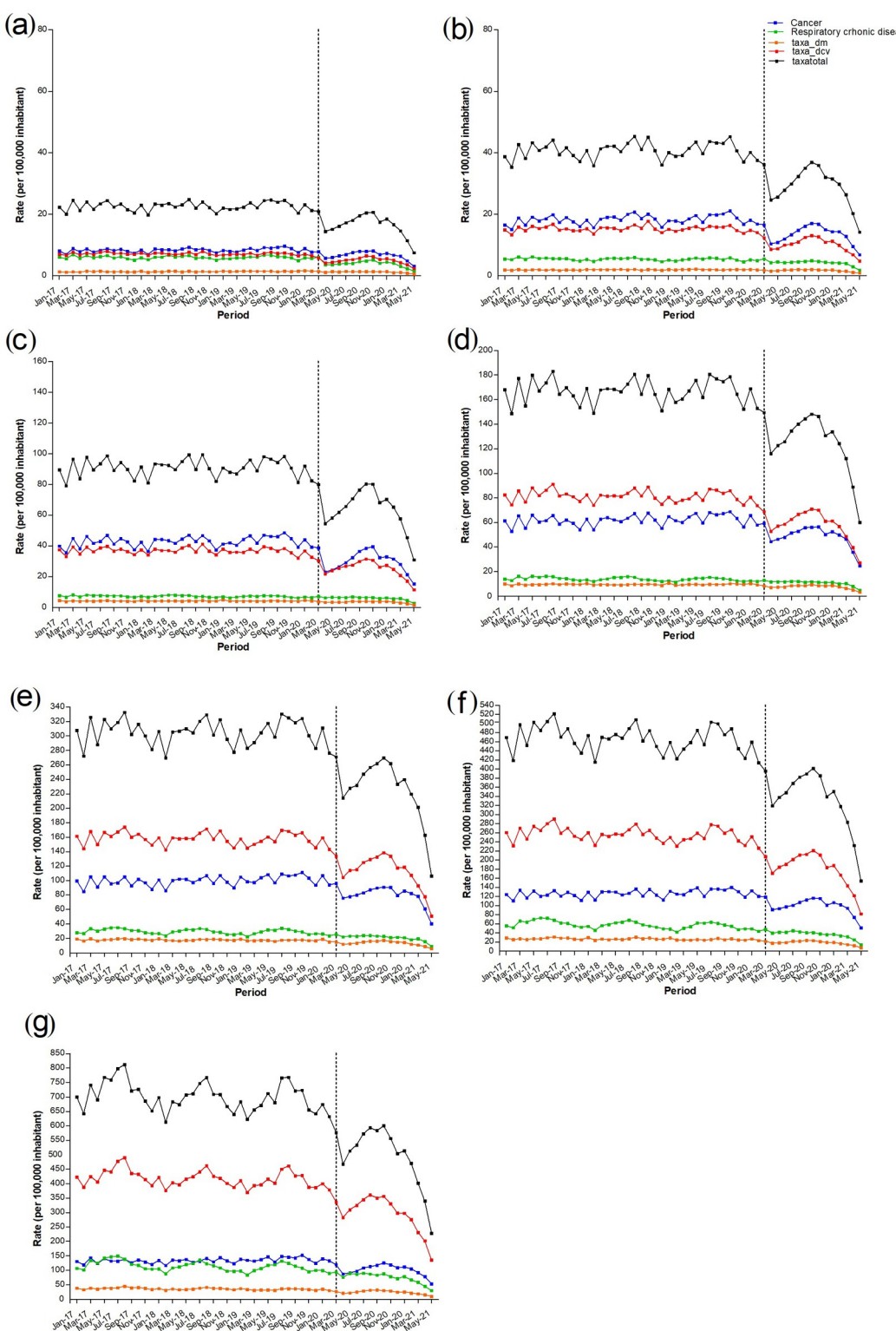

**Fig 3.** Hospital admission rate for chronic non-communicable diseases (NCDs) in Brazil according to the type of NCD and age group: (a) 20–29 years, (b) 30–39 years, (c) 40–49 years, (d) 50–59 years, (e) 60–69 years and (f) 70–79 years and (g) ≥80 years. Pre-pandemic period: January 1, 2017 to February 29, 2020; Pandemic period: March 1, 2020 to May 31, 2021.

**Table 3. Hospital admissions for chronic non-communicable diseases (NCDs) in Brazil before and after the onset of the coronavirus disease pandemic according to the type of NCD and age group.**

| Variables | 20–39 Pre-pandemic Mean | 20–39 Pre-pandemic 95% CI | 20–39 Pandemic Mean | 20–39 Pandemic 95% CI | 20–39 Δ-% (95% CI) | 20–39 p-value* | 40–49 Pre-pandemic Mean | 40–49 Pre-pandemic 95% CI | 40–49 Pandemic Mean | 40–49 Pandemic 95% CI | 40–49 Δ-% (95% CI) | 40–49 p-value* | 50–59 Pre-pandemic Mean | 50–59 Pre-pandemic 95% CI | 50–59 Pandemic Mean | 50–59 Pandemic 95% CI | 50–59 Δ-% (95% CI) | 50–59 p-value* |
|---|---|---|---|---|---|---|---|---|---|---|---|---|---|---|---|---|---|---|
| **Cancer** | | | | | | | | | | | | | | | | | | |
| All cases, sum | 345,809 | | 102,803 | | | | 459,399 | | 132,799 | | | | 541,058 | | 174,058 | | | |
| Cases | 2,275 | 2,035–2,514 | 1,713 | 1,434–1,999 | -24.7 (-29.5;-20.5) | 0.009 | 6,044 | 5,256–6,833 | 4426 | 3,484–5,368 | -26.8 (-33.7;-21.5) | 0.021 | 7,119 | 6,788–7,449 | 5,801 | 5,272–6,330 | -18.5 (-22.3;-15.0) | <0.001 |
| Rate (per 100,000) | 13.3 | 11.9–14.7 | 10.0 | 8.4–11.6 | -24.8 (-29.4;-21.1) | 0.008 | 42.5 | 37.0–47.9 | 29.8 | 23.5–36.0 | -29.9 (-36.5;-24.9) | 0.009 | 61.7 | 59.3–64.0 | 48.4 | 44.4–52.5 | -21.6 (-25.1;-17.9) | <0.001 |
| **Diabetes mellitus** | | | | | | | | | | | | | | | | | | |
| All cases, sum | 41,955 | | 14,490 | | | | 44,496 | | 15,027 | | | | 83,697 | | 27,205 | | | |
| Cases | 276 | 265–286 | 242 | 222–260 | -12.3 (-16.2;-9.1) | <0.001 | 585 | 571–599 | 500 | 458–543 | -14.5 (-19.8;-9.3) | <0.001 | 1,101 | 1,070–1,132 | 906 | 811–1,002 | -17.7 (-24.2;-11.5) | <0.001 |
| Rate (per 100,000) | 1.6 | 1.6–1.7 | 1.4 | 1.3–1.5 | -12.5 (-18.7;-11.8) | <0.001 | 4.2 | 4.0–4.3 | 3.4 | 3.0–3.7 | -19.1 (-25.0;-13.9) | <0.001 | 9.6 | 9.3–10.0 | 7.7 | 6.8–8.5 | -19.8 (-26.9;-15.0) | <0.001 |
| **Cardiovascular diseases** | | | | | | | | | | | | | | | | | | |
| All cases, sum | 290,884 | | 76,870 | | | | 394,728 | | 110,879 | | | | 710,289 | | 207,358 | | | |
| Cases | 1,913 | 1,788–2,039 | 1,281 | 1,137–1,425 | -33.0 (-36.4;-30.1) | <0.001 | 5,193 | 5,085–5,301 | 3,695 | 3,397–3,994 | -28.9 (-33.2;-24.7) | <0.001 | 9,345 | 9,121–9,570 | 6,911 | 6,213–7,610 | -26.1 (-31.9;-20.5) | <0.001 |
| Rate (per 100,000) | 11.2 | 10.5–11.9 | 7.5 | 6.4–8.3 | -33.0 (-39.0;-30.2) | <0.001 | 36.8 | 36.1–37.4 | 25.1 | 23.0–27.1 | -31.8 (-36.3;-27.5) | <0.001 | 81.8 | 79.2–84.4 | 58.4 | 52.0–64.9 | -28.6 (-34.4;-23.1) | <0.001 |
| **Chronic respiratory diseases** | | | | | | | | | | | | | | | | | | |
| All cases, sum | 146,322 | | 41,541 | | | | 76,984 | | 26,192 | | | | 121,697 | | 38,810 | | | |
| Cases | 962 | 948–977 | 692 | 646–738 | -28.1 (-31.8;-24.5) | <0.001 | 1013 | 995–1,030 | 873 | 805–940 | -13.8 (-19.1;-8.7) | <0.001 | 1601 | 1,565–1,637 | 1293 | 1,196–1,391 | -19.2 (-23.6;-15.0) | <0.001 |
| Rate (per 100,000) | 5.6 | 5.5–5.7 | 4.0 | 3.8–4.3 | -29.0 (-30.9;-24.7) | <0.001 | 7.2 | 7.1–7.3 | 5.9 | 5.4–6.4 | -18.1 (-23.9;-12.3) | <0.001 | 14.1 | 13.6–14.3 | 10.9 | 10.0–11.8 | -22.7 (-26.5;-17.5) | <0.001 |
| **All types** | | | | | | | | | | | | | | | | | | |
| All cases, sum | 824,970 | | 235,704 | | | | 975,607 | | 284,897 | | | | 1,456,741 | | 447,431 | | | |
| Cases | 5,327 | 5,070–5,783 | 3,928 | 3498–4357 | -26.3 (-31.0;-24.7) | <0.001 | 12,836 | 11,962–13,710 | 9,496 | 8,416–10,576 | -26.0 (-29.7;-22.9) | <0.001 | 19,167 | 18,888–19,446 | 14,914 | 13,839–15,988 | -22.2 (-26.7;-17.8) | <0.001 |
| Rate (per 100,000) | 31.7 | 29.6–33.8 | 22.9 | 20.4–25.4 | -27.8 (-31.1;-24.9) | <0.001 | 90.6 | 84.7–96.4 | 64.2 | 57.1–71.2 | -29.0 (-32.6;-26.1) | <0.001 | 167.1 | 164.8–169.4 | 125.5 | 115.9–135.0 | -24.9 (-29.7;-20.3) | <0.001 |

(*Continued*)

**Table 3.** (Continued)

| Variables | 60–69 years | | | | | | 70–79 years | | | | | | ≥80 years | | | | | |
|---|---|---|---|---|---|---|---|---|---|---|---|---|---|---|---|---|---|---|
| | Pre-pandemic | | Pandemic | | Δ - % (95.0% CI) | p-value* | Pre-pandemic | | Pandemic | | Δ - % (95.0% CI) | p-value* | Pre-pandemic | | Pandemic | | Δ - % (95.0% CI) | p-value* |
| | Mean | 95.0% CI | Mean | 95.0% CI | | | Mean | 95.0% CI | Mean | 95.0% CI | | | Mean | 95.0% CI | Mean | 95.0% CI | | |
| **Cancer** | | | | | | | | | | | | | | | | | | |
| All cases, sum | 584,197 | | 199,633 | | | | 380,666 | | 128,898 | | | | 160,070 | | 52,269 | | | |
| Cases | 7,686 | 7,525–7,848 | 6,654 | 6,226–7,082 | -13.4 (-17.3;-9.8) | <0.001 | 5,008 | 4,859–5,158 | 4,296 | 3,967–4,626 | -14.2 (-18.3;-10.3) | <0.001 | 2,106 | 2,061–2,150 | 1,742 | 1,621–1,863 | -17.3 (-21.4;-13.3) | <0.001 |
| Rate (per 100,000) | 100.6 | 97.7–103.5 | 80.5 | 74.3–86.7 | -20.0 (-23.9;-16.2) | <0.001 | 130.5 | 123.4–137.6 | 101.8 | 90.3–113.2 | -22.0 (-26.9;-17.7) | <0.001 | 142.5 | 133.8–151.2 | 108.4 | 95.1–121.7 | -24.0 (-28.9;-19.5) | <0.001 |
| **diabetes mellitus** | | | | | | | | | | | | | | | | | | |
| All cases, sum | 104,519 | | 33,587 | | | | 79,643 | | 23,971 | | | | 42,617 | | 12,347 | | | |
| Cases | 1,375 | 1,348–1,401 | 1,119 | 1,008–1,230 | -18.6 (-25.2;-12.2) | <0.001 | 1047 | 1,023–1,072 | 799 | 729–868 | -23.7 (-28.8;-19.0) | <0.001 | 560 | 531–590 | 411 | 365–457 | -26.6 (-31.6;-22.5) | <0.001 |
| Rate (per 100,000) | 18.0 | 17.5–18.6 | 13.6 | 12.0–15.2 | -24.4 (-31.5;-18.3) | <0.001 | 26.7 | 26.1–27.3 | 18.7 | 16.0–20.7 | -30.0 (-38.7;-24.2) | <0.001 | 35.9 | 35.1–36.8 | 24.5 | 22.1–26.9 | -31.8 (-33.0;-26.9) | <0.001 |
| **Cardiovascular diseases** | | | | | | | | | | | | | | | | | | |
| All cases, sum | 925,606 | | 281,820 | | | | 768,922 | | 234,726 | | | | 496,750 | | 151,558 | | | |
| Cases | 12,179 | 11,786–12,571 | 9,394 | 8,440–10,347 | -22.9 (-28.4;-17.7) | <0.001 | 10117 | 9,929–10,305 | 7,824 | 7,187–8,460 | -22.7 (-27.6;-17.9) | <0.001 | 6,536 | 6,327–6,744 | 5,051 | 4,603–5,500 | -22.7 (-27.2;-18.5) | <0.001 |
| Rate (per 100,000) | 160.5 | 153.0–168.0 | 114.7 | 100.8–128.6 | -28.5 (-34.1;-23.5) | <0.001 | 262.3 | 250.7–273.9 | 184.5 | 164.1–205.1 | -29.7 (-34.5;-25.1) | <0.001 | 427.4 | 414.9–439.9 | 303.6 | 276.2–331.1 | -29.9 (-33.5;-24.7) | <0.001 |
| **Chronic respiratory diseases** | | | | | | | | | | | | | | | | | | |
| All cases, sum | 171,419 | | 52,414 | | | | 171,717 | | 48,079 | | | | 135,077 | | 38,358 | | | |
| Cases | 2,255 | 2,194–2,316 | 1,747 | 1,606–1,887 | -22.5 (-26.8;-18.5) | <0.001 | 2,259 | 2,190–2,328 | 1,602 | 1,463–1,741 | -29.1 (-33.2;-25.2) | <0.001 | 1,777 | 1,720–1,834 | 1,278 | 1,167–1,389 | -28.1 (-32.1;-24.3) | <0.001 |
| Rate (per 100,000) | 29.6 | 28.5–30.7 | 21.2 | 19.1–23.3 | -28.4 (-32.9;-24.1) | <0.001 | 58.7 | 55.6–61.8 | 37.8 | 33.4–42.2 | -35.6 (-39.9;-31.7) | <0.001 | 118.5 | 112.1–124.9 | 78.4 | 69.1–87.7 | -33.8 (-38.4;-29.8) | <0.001 |
| **All types** | | | | | | | | | | | | | | | | | | |
| All cases, sum | 1,785,741 | | 567,454 | | | | 1,400,948 | | 435,674 | | | | 834,514 | | 254,532 | | | |
| Cases | 23,496 | 22,920–24,072 | 18,915 | 17,334–20,496 | -19.5 (-24.4;-14.9) | <0.001 | 18,433 | 18,068–18,798 | 14,522 | 13,384–15,660 | -21.2 (-25.9;-16.7) | <0.001 | 10,980 | 10,705–11,255 | 8,484 | 7,806–9,162 | -22.7 (-27.1;-18.6) | <0.001 |
| Rate (per 100,000) | 308.7 | 297.1–320.4 | 230.0 | 206.6–253.5 | -25.4 (-30.5;-20.9) | <0.001 | 478.3 | 456.5–500.1 | 342.9 | 305.2–380.6 | -28.2 (-33.1;-23.9) | <0.001 | 724.4 | 697.6–751.3 | 515.0 | 464.7–565.3 | -28.7 (-33.4;-24.7) | <0.001 |

Note: The mean number and rate of hospital admissions were compared between the pre-pandemic (January 1, 2017 to February 29, 2020) and pandemic (March 1, 2020 to May 31, 2021) periods. 95.0% CI = 95.0% confidence interval; *Student's t-test for independent samples.

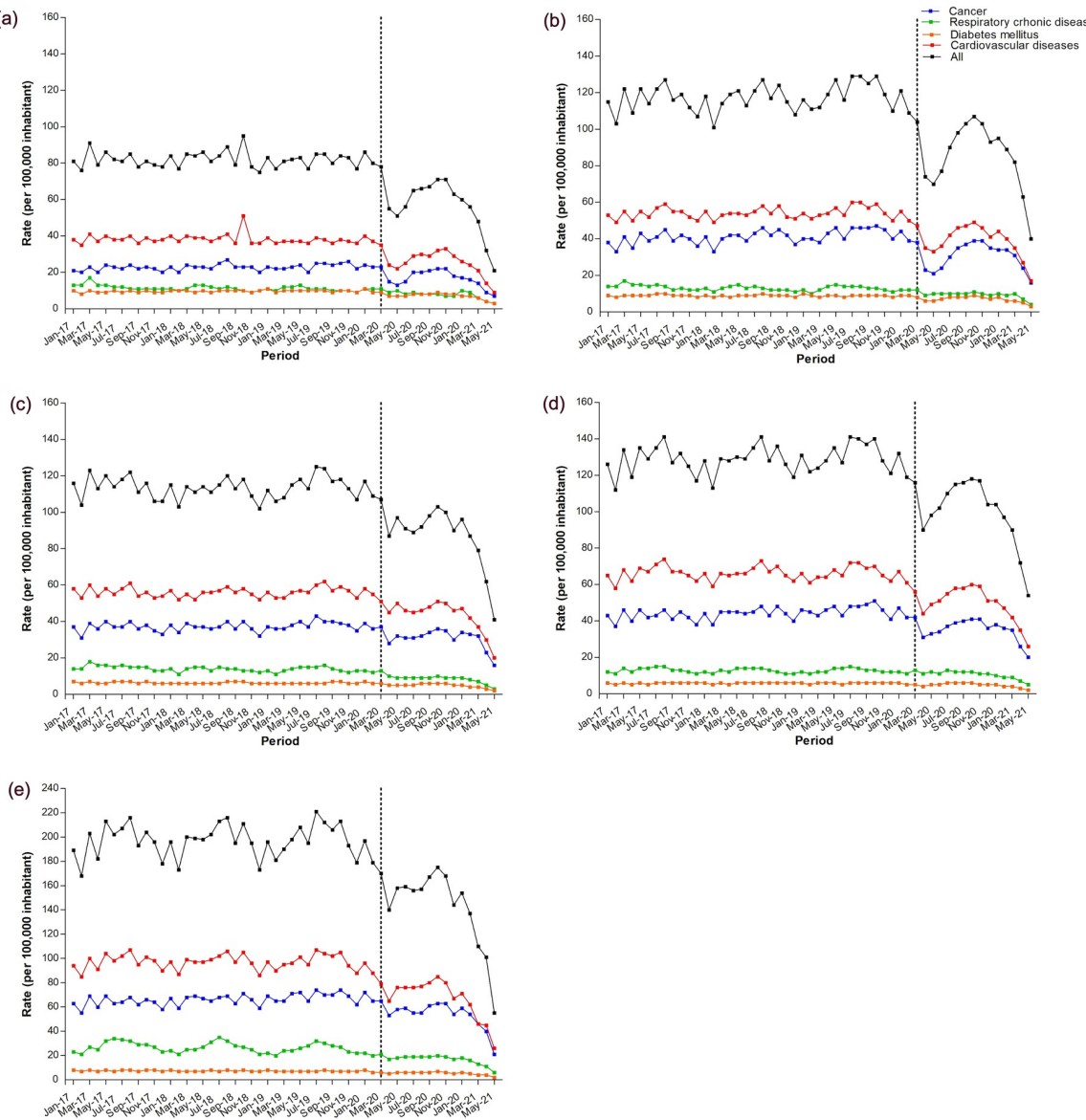

**Fig 4.** Hospital admission rate for chronic non-communicable diseases (NCDs) in Brazil according to the type of NCD and region of Brazil: (a) North, (b) Northeast, (c) Midwest, (d) Southeast, and (e) South. Pre-pandemic period: January 1, 2017 to February 29, 2020; Pandemic period: March 1, 2020 to May 31, 2021.

greater decline in CVD-related hospital admissions than in cancer-related (30.0% vs. 23.0%) and diabetes mellitus-related (30.0% vs. 24.0%) hospital admissions during the pandemic period from the pre-pandemic period.

Table 6 presents the results of analyses stratified by sex, age group, and region of Brazil for hospital admissions. There was a decline in the number of hospital admissions. regardless of age group, sex, and region of Brazil (Table 6). Similar results were verified for causes related to cancer (Table 7), diabetes mellitus (Table 8), CVD (Table 9), and CRD (Table 10). In addition, there was a greater decrease for most of causes related to NCDs in the Northern and Southern regions of the country than in other regions (Tables 6–10).

**Table 4. Hospital admissions for chronic non-communicable diseases (NCDs) in Brazil before and after the onset of the coronavirus disease pandemic according to the type of NCD and region of Brazil.**

| Variables | North Pre-pandemic Mean | North Pre-pandemic 95.0% CI | North Pandemic Mean | North Pandemic 95.0% CI | North Δ - % (95.0%CI) | North p-value* | Midwest Pre-pandemic Mean | Midwest Pre-pandemic 95.0% CI | Midwest Pandemic Mean | Midwest Pandemic 95.0% CI | Midwest Δ - % (95.0%CI) | Midwest p-value* |
|---|---|---|---|---|---|---|---|---|---|---|---|---|
| **Cancer** | | | | | | | | | | | | |
| All cases, sum | 98,620 | | 29,741 | | | | 159,107 | | 54,498 | | | |
| Cases | 185 | 172–197 | 141 | 125–157 | -23.8(-27.3;-20.3) | <0.001 | 1,130 | 1,065–1,196 | 867 | 786–948 | -23.4(-26.2;-20.7) | <0.001 |
| Rate (per 100,000) | 40.8 | 38.2–43.3 | 28.9 | 25.8–32.1 | -29.2(-32.5;-25.9) | <0.001 | 65.3 | 61.3–69.2 | 48.3 | 43.4–53.3 | -26.0(-29.2;-22.9) | <0.001 |
| **Diabetes mellitus** | | | | | | | | | | | | |
| All cases, sum | 41,296 | | 12,630 | | | | 26,867 | | 8,307 | | | |
| Cases | 77 | 73–81 | 60 | 54–65 | -22.1(-26.0;-19.7) | <0.001 | 242 | 230–254 | 190 | 175–206 | -21.5(-23.9;-18.9) | <0.001 |
| Rate (per 100,000) | 26.8 | 24.5–29.0 | 18.1 | 15.6–20.5 | -32.5(-36.3;-29.3) | <0.001 | 20.7 | 18.9–22.5 | 14.5 | 12.5–16.6 | -30.0(-33.9;-26.2) | <0.001 |
| **Cardiovascular diseases** | | | | | | | | | | | | |
| All cases, sum | 163,891 | | 45,477 | | | | 239,092 | | 76,571 | | | |
| Cases | 308 | 294–321 | 216 | 198–235 | -29.8(-32.6;-26.8) | <0.001 | 1484 | 1,418–1,550 | 1101 | 1,011–1,190 | -25.8(-28.7;-23.3) | <0.001 |
| Rate (per 100,000) | 128.9 | 115.4–142.4 | 79.3 | 65.8–92.9 | -38.5(-42.9;-34.8) | <0.001 | 135.7 | 122.6–148.7 | 93.0 | 78.4–107.5 | -31.5(-30.0;-27.7) | <0.001 |
| **Chronic respiratory diseases** | | | | | | | | | | | | |
| All cases, sum | 48,626 | | 14,043 | | | | 59,518 | | 15,151 | | | |
| Cases | 91 | 89–93 | 66 | 63–70 | -27.5(-29.2;-24.7) | <0.001 | 358 | 349–367 | 270 | 258–282 | -24.6(-26.1;-23.2) | <0.001 |
| Rate (per 100,000) | 36.8 | 32.6–41.0 | 22.7 | 18.6–27.0 | -38.2(-42.9;-34.1) | <0.001 | 32.8 | 29.3–36.2 | 22.2 | 18.5–25.9 | -32.3(-36.9;-28.4) | <0.001 |
| **All types** | | | | | | | | | | | | |
| All cases, sum | 352,433 | | 101,891 | | | | 484,584 | | 154,527 | | | |
| Cases | 662 | 638–685 | 485 | 451–519 | -26.7(-29.3;-24.2) | <0.001 | 910 | 873–947 | 735 | 682–788 | -19.2(-21.9;-16.8) | <0.001 |
| Rate (per 100,000) | 233.2 | 211.4–255.1 | 149.1 | 126.6–171.6 | -36.1(-40.1;-32.7) | <0.001 | 259.6 | 237.5–281.6 | 188.7 | 162.4–215.0 | -27.1(-31.6;-23.6) | <0.001 |

| Variables | Northeast Pre-pandemic Mean | Northeast Pre-pandemic 95.0% CI | Northeast Pandemic Mean | Northeast Pandemic 95.0% CI | Northeast Δ - % (95.0% CI) | Northeast p-value* | South Pre-pandemic Mean | South Pre-pandemic 95.0% CI | South Pandemic Mean | South Pandemic 95.0% CI | South Δ - % (95.0% CI) | South p-value* | Southeast Pre-pandemic Mean | Southeast Pre-pandemic 95.0% CI | Southeast Pandemic Mean | Southeast Pandemic 95.0% CI | Southeast Δ - % (95.0%CI) | Southeast p-value* |
|---|---|---|---|---|---|---|---|---|---|---|---|---|---|---|---|---|---|---|
| **Cancer** | | | | | | | | | | | | | | | | | | |
| All cases, sum | 601,667 | | 182,129 | | | | 545,857 | | 179,695 | | | | 1,065,948 | | 344,397 | | | |
| Cases | 1,130 | 1,065–1,196 | 867 | 786–948 | -23.3(-26.2;-20.8) | <0.001 | 1026 | 973–1,079 | 855 | 779–932 | -16.7(-19.9;-13.6) | <0.001 | 2003 | 1,899–2,107 | 1,639 | 1,498–1,780 | -18.2(-21.1;-15.5) | <0.001 |
| Rate (per 100,000) | 65.3 | 61.3–69.2 | 48.3 | 43.4–53.3 | -26.0(-29.2;-22.9) | <0.001 | 101.7 | 95.2–108.4 | 80.4 | 71.6–89.2 | -20.9(-24.8;-17.7) | <0.001 | 69.5 | 64.9–74.0 | 53.9 | 48.0–59.7 | -22.4(-26.0;-19.3) | <0.001 |
| **Diabetes mellitus** | | | | | | | | | | | | | | | | | | |
| All cases, sum | 129,187 | | 40,093 | | | | 60,540 | | 18,113 | | | | 139,037 | | 47,484 | | | |
| Cases | 242 | 230–254 | 190 | 175–206 | -21.5(-23.9;-18.9) | <0.001 | 113 | 108–119 | 86 | 79–93 | -23.9(-26.8;-21.8) | <0.001 | 261 | 248–273 | 226 | 206–245 | -13.4(-16.9;-10.2) | 0.003 |
| Rate (per 100,000) | 20.7 | 18.9–22.5 | 14.5 | 12.5–16.6 | -30.0(-33.9;-22.2) | <0.001 | 12.8 | 11.8–13.7 | 8.7 | 7.7–9.8 | -32.0(-34.7;-28.5) | <0.001 | 9.7 | 9.1–10.4 | 7.7 | 6.7–8.6 | -20.6(-26.4;-17.3) | 0.001 |

*(Continued)*

**Table 4.** (Continued)

| | Group 1 Pre | 95% CI | Group 1 Pandemic | 95% CI | Change (95% CI) | p | Group 2 Pre | 95% CI | Group 2 Pandemic | 95% CI | Change (95% CI) | p | Group 3 Pre | 95% CI | Group 3 Pandemic | 95% CI | Change (95% CI) | p |
|---|---|---|---|---|---|---|---|---|---|---|---|---|---|---|---|---|---|---|
| **Cardiovascular diseases** | | | | | | | | | | | | | | | | | | |
| All cases, sum | 789,764 | | 231,271 | | | | 799,934 | | 225,495 | | | | 1,594,475 | | 484,397 | | | |
| Cases | 1,484 | 1,418–1,550 | 1,101 | 1,011–1,190 | -25.8 (-28.7; -23.2) | <0.001 | 1,503 | 1,422–1,584 | 1,073 | 967–1,179 | -28.6 (-31.9; -25.6) | <0.001 | 2,997 | 2,848–3,146 | 2,306 | 2,099–2,513 | -23.1(-26.3; -20.1) | <0.001 |
| Rate (per 100,000) | 135.7 | 122.6–148.7 | 93.0 | 78.4–107.5 | -31.5 (-36.0; -27.7) | <0.001 | 190.4 | 174.1–206.8 | 129.8 | 110.6–149.0 | -31.8 (-36.5; -27.9) | <0.001 | 129.7 | 118.8–140.6 | 95.5 | 81.9–109.0 | -26.4(-31.1; -22.5) | <0.001 |
| **Chronic respiratory diseases** | | | | | | | | | | | | | | | | | | |
| All cases, sum | 190,591 | | 56,790 | | | | 217,907 | | 56,485 | | | | 306,597 | | 102,925 | | | |
| Cases | 358 | 349–367 | 270 | 258–282 | -24.6 (-26.1; -23.2) | <0.001 | 409 | 392–427 | 268 | 249–288 | -34.5 (-36.5; -32.6) | <0.001 | 576 | 557–594 | 490 | 460–519 | -14.9(-17.5; -12.6) | <0.001 |
| Rate (per 100,000) | 32.8 | 29.3–36.2 | 22.2 | 18.5–25.9 | -32.3 (-36.7; -28.5) | <0.001 | 55.3 | 49.8–60.8 | 32.7 | 27.6–37.4 | -40.9 (-44.6; -38.5) | <0.001 | 26.7 | 24.1–29.3 | 20.7 | 17.5–23.8 | -22.5(-27.4; -18.8) | 0.010 |
| **All types** | | | | | | | | | | | | | | | | | | |
| All cases, sum | 1,711,209 | | 510,283 | | | | 1,624,238 | | 479,788 | | | | 3,106,057 | | 979,203 | | | |
| Cases | 3,216 | 3,097–3,335 | 2,429 | 2,264–2,595 | -24.5 (-26.9; -22.2) | <0.001 | 3053 | 2,905–3,200 | 2,284 | 2,085–2,483 | -25.2 (-28.2; -22.4) | <0.001 | 5,838 | 5,572–6,104 | 4,662 | 4,288–5,037 | -20.1(-23.0; -17.5) | <0.001 |
| Rate (per 100,000) | 254.4 | 232.8–276.0 | 178.0 | 153.6–202.6 | -30.0 (-34.0; -26.6) | <0.001 | 360.3 | 331.6–388.9 | 251.7 | 218.3–285.1 | -30.1 (-34.2; -26.7) | <0.001 | 235.6 | 217.2–254.0 | 177.7 | 154.8–200.6 | -24.6(-28.7; -21.0) | <0.001 |

Note: The mean number and rate of hospital admissions were compared between the pre-pandemic (January 1, 2017 to February 29, 2020) and pandemic (March 1, 2020 to May 31, 2021) periods. 95.0% CI = 95.0% confidence interval; *Student's t-test for independent samples.

**Table 5. Poisson multiple regression models of the impact of the coronavirus disease pandemic on hospital admissions for chronic non-communicable diseases (NCDs) in Brazil according to the type of NCDs.**

| Type of NCDs | IRR | 95.0% CI | β | p-value* |
|---|---|---|---|---|
| Cancer | | | | |
| Pre-pandemic | Reference | | | |
| Pandemic | 0.77 | 0.74–0.79 | -0.265 | <0.001 |
| Chronic respiratory diseases | | | | |
| Pre-pandemic | Reference | | | |
| Pandemic | 0.71 | 0.70–0.73 | -0.342 | <0.001 |
| Diabetes mellitus | | | | |
| Pre-pandemic | Reference | | | |
| Pandemic | 0.76 | 0.74–0.78 | -0.276 | <0.001 |
| Cardiovascular diseases | | | | |
| Pre-pandemic | Reference | | | |
| Pandemic | 0.70 | 0.69–0.72 | -0.341 | <0.001 |
| All types of NCD | | | | |
| Pre-pandemic | Reference | | | |
| Pandemic | 0.73 | 0.71–0.75 | -0.316 | <0.001 |

Note: Each Poisson multiple regression model was adjusted for sex, age group, region of Brazil, month, and the dummy variable indicating the impact of the pandemic. The resident population was included as an exposure variable. Pre-pandemic period: January 1, 2017 to February 29, 2020; Pandemic period: March 1, 2020 to May 31, 2021. IRR = incidence rate ratio; 95.0% CI = 95.0% confidence interval; β = regression coefficient; *Wald statistic.

## Discussion

This study analyzed the impact of the COVID-19 pandemic on the hospitalization rate for NCDs in Brazil, stratified according to the type of NCDs, sex, age group, and region of Brazil. We found a significant decrease in the number of hospital admissions for all types of NCD after the onset of the COVID-19 pandemic, regardless of sex, age group, and region of Brazil. We also observed the largest declines in NCD-related hospital admissions in the North and South after the onset of the pandemic.

Similar studies conducted in both developing and developed countries have also reported a decrease in the number of hospitalizations for chronic conditions after the onset of the COVID-19 pandemic [13, 17–20, 34, 35]. An investigation conducted in four hospitals in New York, an epicenter of the COVID-19 pandemic in the United States, showed that weekly hospitalizations increased by 144.0% at the peak of the pandemic, while a decrease was observed in hospitalizations for exacerbations of chronic conditions such as heart failure and chronic obstructive pulmonary disease (COPD) [34]. Another study conducted in the United States that included data from 201 hospitals in 36 states found a decline in hospital admissions for diabetes (35.8% decrease), congestive heart failure (43.8% decrease), COPD or asthma (61.6% decrease), and other chronic conditions during the pandemic period (April 2020) compared to those during the pre-pandemic period [13]. In Alberta, Canada, an investigation showed a decrease in emergency admissions related to chronic conditions during the pandemic period —diabetes mellitus (21.0% decrease), COPD (25.0% decrease), and arterial hypertension (29.0% decrease) [18]. A study in Hong Kong, China, estimated a 44.0% decrease in the number of hospital admissions related to COPD [19].

Other studies in Brazil have also reported similar results [16, 17, 20, 35, 36]. In a national survey using SIH-SUS data, hospital admission rates for clinical cancer decreased from 13.9 to

**Table 6. Poisson multiple regression models of the impact of the coronavirus disease pandemic on the hospitalization rate for all types of chronic non-communicable diseases in Brazil according to subgroups by age group, sex, and region of Brazil.**

| Variables | IRR | 95.0% CI | β | p-value* |
|---|---|---|---|---|
| **Age group (years)** | | | | |
| 20–39 | 0.72 | 0.68–0.77 | -0.322 | <0.001 |
| 40–49 | 0.70 | 0.68–0.74 | -0.344 | <0.001 |
| 50–59 | 0.75 | 0.72–0.78 | -0.286 | <0.001 |
| 60–69 | 0.74 | 0.72–0.77 | -0.296 | <0.001 |
| 70–79 | 0.71 | 0.69–0.74 | -0.336 | <0.001 |
| ≥80 | 0.70 | 0.68–0.74 | -0.346 | <0.001 |
| **Sex** | | | | |
| Female | 0.76 | 0.74–0.78 | -0.272 | <0.001 |
| Male | 0.70 | 0.68–0.72 | -0.359 | <0.001 |
| **Region of Brazil** | | | | |
| North | 0.68 | 0.64–0.71 | -0.390 | <0.001 |
| Midwest | 0.75 | 0.72–0.78 | -0.293 | <0.001 |
| Northeast | 0.71 | 0.68–0.75 | -0.338 | <0.001 |
| Southeast | 0.75 | 0.72–0.78 | -0.280 | <0.001 |
| South | 0.70 | 0.68–0.84 | -0.349 | <0.001 |

Note: The adjusted variables for the models of each subgroup were sex, age group, region of Brazil, month, and the dummy variable indicating the impact of the pandemic. All these variables were included in each subgroup model, except the dummy variable of each subgroup in the respective model. The resident population was included as an exposure variable. Pre-pandemic period: January 1, 2017 to February 29, 2020; Pandemic period: March 1, 2020 to May 31, 2021. IRR = incidence rate ratio; 95.0% CI = 95.0% confidence interval; β = regression coefficient; *Wald statistic.

10.2 cases per 100,000 inhabitants (26.6% decrease), while the admission rates for surgical cancer decreased from 20.2 to 14.5 cases per 100,000 inhabitants (28.2% decrease) after the onset of the pandemic between March and June 2020 compared to those during the same period prior to the pandemic [17]. Similarly, another study found a 21.0% decrease in the number of cancer-related hospitalizations compared to those in the period in 2020 [20]. An additional study identified a significant decrease in the number of hospital admissions for NCDs, including cancer; CVD; and endocrine, nutritional, and metabolic diseases, between January and June 2020 in Brazil [16]. An investigation showed a 15.0% decline in the hospitalization rate for CVD in Brazil between March and May 2020 compared to that in the same period in 2019, regardless of the age group [35]. Using SIH-SUS data, a study also detected a 6.6% decline in the number of admissions related to diabetes mellitus throughout Brazil in 2020 compared to that in 2019 [36].

The causes for the decline in hospital admissions during the period after the onset of the COVID-19 pandemic are multifactorial. Demand for health services is negatively impacted by fears of becoming infected with SARS-CoV-2 by other patients, mobility restriction policies, social isolation, and reduced access to healthcare during the pandemic, including postponement of appointments and elective care procedures as healthcare resources were redirected to care for patients with COVID-19 [13, 17, 34, 35, 37]. In particular, prevention, early detection and continuous monitoring of patients with NCDs that are carried out mainly in primary health care have suffered a significant reduction, impacting access to health services and hospitalizations [38]. Brazil, due to the scarcity of health resources and the pressure under the

**Table 7. Poisson multiple regression models of the impact of the coronavirus disease pandemic on the hospitalization rate for cancer in Brazil according to subgroups by age group, sex, and region of Brazil.**

| Variables | IRR | 95.0% CI | β | p-value* |
|---|---|---|---|---|
| **Age group (years)** | | | | |
| 20–39 | 0.75 | 0.70–0.82 | -0.282 | <0.001 |
| 40–49 | 0.70 | 0.66–0.74 | -0.254 | <0.001 |
| 50–59 | 0.79 | 0.75–0.82 | -0.239 | <0.001 |
| 60–69 | 0.80 | 0.77–0.83 | -0.223 | <0.001 |
| 70–79 | 0.78 | 0.75–0.80 | -0.253 | <0.001 |
| ≥80 | 0.76 | 0.73–0.79 | -0.275 | <0.001 |
| **Sex** | | | | |
| Male | 0.80 | 0.78–0.82 | -0.220 | <0.001 |
| Female | 0.74 | 0.72–0.76 | -0.299 | <0.001 |
| **Region of Brazil** | | | | |
| Midwest | 0.80 | 0.76–0.86 | -0.214 | <0.001 |
| Northeast | 0.73 | 0.67–0.79 | -0.317 | <0.001 |
| North | 0.71 | 0.64–0.79 | -0.337 | <0.001 |
| Southeast | 0.77 | 0.74–0.82 | -0.255 | <0.001 |
| South | 0.79 | 0.75–0.83 | -0.236 | <0.001 |

Note: The adjusted variables for the models of each subgroup were sex, age group, region of Brazil, month, and the dummy variable indicating the impact of the pandemic. All these variables were included in each subgroup model, except for the dummy variable of each subgroup in the respective model. The resident population was included as an exposure variable. Pre-pandemic period: January 1, 2017 to February 29, 2020; Pandemic period: March 1, 2020 to May 31, 2021. IRR = incidence rate ratio; 95.0% CI = 95.0% confidence interval; β = regression coefficient; *Wald statistic.

system, shifted the beds for the care of patients with NCDs to patients with COVID-19, impacting on the reduction of hospitalizations of patients [38]. In addition, healthcare professionals have been disproportionately affected by the COVID-19 pandemic with high infection rates, leading to reduced capacity to care for patients with NCDs [39, 40].

This study also demonstrated a larger decrease in hospital admissions for most types of NCDs in the North and South of Brazil. The Northern region was the first to experience an increase in the number of cases and a collapse of the healthcare system. The exponential increase in COVID-19 cases led authorities to adopt more proactive, rigorous social distancing measures, which significantly affected the population's access to health services, which may explain the larger decline in hospital admissions for NCDs in the North than in other regions [5, 20]. The Southern region also had a higher incidence of COVID-19 cases per 100,000 inhabitants than all other regions, except the Midwest, and subsequently adopted stricter measures for social distancing and access to health services. Additionally, the Northern region is one of the least developed regions in Brazil and has the least amount of economic resources. Even before the COVID-19 pandemic, individuals from the North had less access to health services compared to other regions [17]. Therefore, our results indicated that individuals with NCDs had unequal access to health services during the COVID-19 pandemic, pointing to the need for strategies to promote greater equity in health access in all regions, especially those with larger gaps in health services.

This study has some limitations. First, the study only included hospitalizations of individuals reported in the SUS and did not include hospitalizations in the private system. Although the SIH-SUS covers 70.0% of Brazilian hospitalizations [41], the non-inclusion of

**Table 8. Poisson multiple regression models of the impact of the coronavirus disease pandemic on the hospitalization rate for diabetes mellitus in Brazil according to subgroups by age group, sex, and region of Brazil.**

| Variables | IRR | 95.0% CI | β | p-value* |
|---|---|---|---|---|
| **Age group (years)** | | | | |
| 20–39 | 0.87 | 0.84–0.91 | -0.133 | <0.001 |
| 40–49 | 0.82 | 0.78–0.85 | -0.204 | <0.001 |
| 50–59 | 0.79 | 0.75–0.83 | -0.235 | <0.001 |
| 60–69 | 0.75 | 0.71–0.79 | -0.285 | <0.001 |
| 70–79 | 0.70 | 0.66–0.73 | -0.364 | <0.001 |
| ≥80 | 0.67 | 0.64–0.71 | -0.395 | <0.001 |
| **Sex** | | | | |
| Male | 0.81 | 0.79–0.84 | -0.207 | <0.001 |
| Female | 0.70 | 0.68–0.73 | -0.350 | <0.001 |
| **Region of Brazil** | | | | |
| Midwest | 0.72 | 0.69–0.77 | -0.322 | <0.001 |
| Northeast | 0.74 | 0.70–0.77 | -0.302 | <0.001 |
| North | 0.71 | 0.68–0.75 | -0.339 | <0.001 |
| Southeast | 0.82 | 0.77–0.87 | -0.200 | <0.001 |
| South | 0.71 | 0.67–0.75 | -0.337 | <0.001 |

Note: The adjusted variables for the models of each subgroup were sex, age group, region of Brazil, month, and the dummy variable indicating the impact of the pandemic. All these variables were included in each subgroup model, except the dummy variable of each subgroup in the respective model. The resident population was included as an exposure variable. Pre-pandemic period: January 1, 2017 to February 29, 2020; Pandemic period: March 1, 2020 to May 31, 2021. IRR = incidence rate ratio; 95.0% CI = 95.0% confidence interval; β = regression coefficient; *Wald statistic.

hospitalizations in the private system may have led to underestimation of hospitalization rates. As these data were not included in our analysis of the impact of the pandemic on the hospitalization rate for NCDs, the results of this study may be under- or overestimated. Second, individuals in the private system may present different characteristics from those hospitalized in the public system (e.g., distribution by age, race and sex). The private sector in Brazil has a higher proportion of female, older, white and more educated users, therefore with a higher socioeconomic level when compared to the public sector (Unified Health System) [42]. The private sector encompasses a vast diversification of hospital services and an even greater concentration of revenue compared to the public sector. Therefore, it would be impossible to generalize the results of the study for the entire Brazilian population and these differences, if analyzed, may impact the estimates. Third, there was a high proportion of missing data for many variables in dataset, such as race, and education level, which made it impossible to use these variables in the present study to analyze the impact according with these variables. Thus, it was impossible to analyze the sensitivity of the pandemic's impact according to these variables. Fourth, this study did not analyze the specific CID-10 in each NCD subgroup. The pandemic's impact may differ for specific groups of diseases and chronic conditions (e.g., hypertension, specific cancers, and COPD). Five, important confounders were not included in the analysis, such as the number of hospital beds, number of healthcare workers per inhabitants, degree of social distancing measures in the regions, including specific restrictions and prevalence of NCDs in the regions, for example. Sixth, we do not have data on the prevalence and characteristics of patients with NCDs before and during the COVID-19 pandemic, according to a large group of causes, which makes it difficult to analyze that the results of this

**Table 9. Poisson multiple regression models of the impact of the coronavirus disease pandemic on the hospitalization rate for cardiovascular disease in Brazil according to subgroups by age group, sex, and region of Brazil.**

| Variables | IRR | 95.0% CI | β | p-value* |
|---|---|---|---|---|
| **Age group (years)** | | | | |
| 20–39 | 0.67 | 0.63–0.72 | -0.400 | <0.001 |
| 40–49 | 0.68 | 0.65–0.71 | -0.382 | <0.001 |
| 50–59 | 0.71 | 0.69–0.75 | -0.336 | <0.001 |
| 60–69 | 0.71 | 0.68–0.74 | -0.339 | <0.001 |
| 70–79 | 0.70 | 0.67–0.73 | -0.354 | <0.001 |
| ≥80 | 0.70 | 0.68–0.74 | -0.343 | <0.001 |
| **Sex** | | | | |
| Male | 0.74 | 0.72–0.75 | -0.304 | <0.001 |
| Female | 0.67 | 0.65–0.69 | -0.404 | <0.001 |
| **Region of Brazil** | | | | |
| Midwest | 0.75 | 0.71–0.78 | -0.294 | <0.001 |
| Northeast | 0.70 | 0.66–0.73 | -0.361 | <0.001 |
| North | 0.65 | 0.61–0.68 | -0.435 | <0.001 |
| Southeast | 0.73 | 0.69–0.76 | -0.323 | <0.001 |
| South | 0.67 | 0.63–0.70 | -0.400 | <0.001 |

Note: The adjusted variables for the models of each subgroup were sex, age group, region of Brazil, month, and the dummy variable indicating the impact of the pandemic. All these variables were included in each subgroup model, except the dummy variable of each subgroup in the respective model. The resident population was included as an exposure variable. Pre-pandemic period: January 1, 2017 to February 29, 2020; Pandemic period: March 1, 2020 to May 31, 2021. IRR = incidence rate ratio; 95.0% CI = 95.0% confidence interval; β = regression coefficient; *Wald statistic.

study are due only to social distancing measures. Finally, this study only included hospital admissions and not admissions to other levels of care (eg, Primary Health Care).

However, our study also has several strengths. First, our study analyzed the impact of COVID-19 on hospital admissions for all major types of NCDs, including cancer, diabetes mellitus, CVD, and CRD, and was not limited to one group of diseases. We also assessed the impact of the pandemic on subgroups by sex, age, and region of Brazil. We utilized data from Brazil's largest database of hospital admissions throughout all regions of Brazil to help us understand how the COVID-19 pandemic has affected hospital admissions for NCDs in the country. Thus, this study provides important results regarding the changing trends in the use of health services in Brazil, which can support interventions and public policies to mitigate the effects of the pandemic on the care of people with NCDs. Moreover, we considered seasonal variations in the regression models to control for the influence of seasonality on the outcomes.

In conclusion, there was a decrease in the hospitalization rate for NCDs in Brazil during the COVID-19 pandemic in a scenario of social distancing measures and overload of health services. These results are alarming as this decline can result in increased incidence of disability and mortality and may reduce the quality of life of people with NCDs, thus having an immense impact on public health. Multiple NCDs-related conditions require immediate assistance to prevent harm to patients; consequently, fewer hospitalizations can lead to irreversible disabilities. Despite the pandemic, it is important for individuals with NCDs to receive proper care in the hospital system. System managers and government agencies must implement strategies that increase access to hospital services in Brazil during the pandemic period, regardless of the region of Brazil. Actions such as increasing the number of elective procedures, periodic

**Table 10. Poisson multiple regression models of the impact of the coronavirus disease pandemic on the hospitalization rate for chronic respiratory diseases in Brazil according to subgroups by age group, sex, and region of Brazil.**

| Variables | IRR | 95.0% CI | β | p-value* |
|---|---|---|---|---|
| **Age group (years)** | | | | |
| 20–39 | 0.72 | 0.69–0.74 | -0.330 | <0.001 |
| 40–49 | 0.82 | 0.79–0.86 | -0.193 | <0.001 |
| 50–59 | 0.78 | 0.74–0.82 | -0.249 | <0.001 |
| 60–69 | 0.71 | 0.68–0.75 | -0.335 | <0.001 |
| 70–79 | 0.64 | 0.61–0.68 | -0.441 | <0.001 |
| ≥80 | 0.66 | 0.63–0.70 | -0.416 | <0.001 |
| **Sex** | | | | |
| Male | 0.75 | 0.73–0.77 | -0.290 | <0.001 |
| Female | 0.67 | 0.65–0.69 | -0.398 | <0.001 |
| **Region of Brazil** | | | | |
| Midwest | 0.59 | 0.56–0.63 | -0.521 | <0.001 |
| Northeast | 0.71 | 0.69–0.74 | -0.338 | <0.001 |
| North | 0.68 | 0.64–0.71 | -0.391 | <0.001 |
| Southeast | 0.80 | 0.77–0.84 | -0.219 | <0.001 |
| South | 0.62 | 0.59–0.65 | -0.479 | <0.001 |

Note: The adjusted variables for the models of each subgroup were sex, age group, region of Brazil, month, and the dummy variable indicating the impact of the pandemic. All these variables were included in each subgroup model, except the dummy variable of each subgroup in the respective model. The resident population was included as an exposure variable. Pre-pandemic period: January 1, 2017 to February 29, 2020; Pandemic period: March 1, 2020 to May 31, 2021. IRR = incidence rate ratio; 95.0% CI = 95.0% confidence interval; β = regression coefficient; *Wald statistic.

examinations, consultations with specialists and in primary health care, and using telemedicine as an alternative to in-person visits can reduce the risk of hospitalizations for NCDs. Therefore, the findings of this study can contribute to the development, implementation, and effectiveness of public policies and protocols for decision making regarding the management of patients with NCDs in Brazil.

## Author Contributions

**Conceptualization:** Rafael Alves Guimarães, Gabriela Moreira Policena, Raquel Silva Pinheiro, Olavo de Oliveira Braga Neto, Adriana Melo Teixeira, Irisleia Aires Silva, Karla de Aleluia Batista.

**Formal analysis:** Rafael Alves Guimarães.

**Funding acquisition:** Geraldo Andrade de Oliveira.

**Investigation:** Rafael Alves Guimarães, Gabriela Moreira Policena, Hellen da Silva Cintra de Paula, Charlise Fortunato Pedroso, Raquel Silva Pinheiro, Alexander Itria, Olavo de Oliveira Braga Neto, Adriana Melo Teixeira, Irisleia Aires Silva, Karla de Aleluia Batista.

**Methodology:** Rafael Alves Guimarães, Gabriela Moreira Policena.

**Project administration:** Rafael Alves Guimarães, Raquel Silva Pinheiro, Karla de Aleluia Batista.

**Resources:** Geraldo Andrade de Oliveira.

**Supervision:** Rafael Alves Guimarães, Raquel Silva Pinheiro, Karla de Aleluia Batista.

**Validation:** Rafael Alves Guimarães, Raquel Silva Pinheiro, Karla de Aleluia Batista.

**Visualization:** Rafael Alves Guimarães, Raquel Silva Pinheiro, Karla de Aleluia Batista.

**Writing – original draft:** Rafael Alves Guimarães, Gabriela Moreira Policena, Hellen da Silva Cintra de Paula, Charlise Fortunato Pedroso, Alexander Itria.

**Writing – review & editing:** Rafael Alves Guimarães, Raquel Silva Pinheiro, Alexander Itria, Karla de Aleluia Batista.

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
