## [Decision Letter · Decision Letter 0]

6 Jan 2022

PONE-D-21-38179Analysis of the impact of coronavirus disease 19 on hospitalization rates for chronic non-communicable diseases in BrazilPLOS ONE

Dear Dr. Guimarães,

Thank you for submitting your manuscript to PLOS ONE. After careful consideration, we feel that it has merit but does not fully meet PLOS ONE’s publication criteria as it currently stands. Therefore, we invite you to submit a revised version of the manuscript that addresses the points raised during the review process.

Your findings are of interest but there are important areas for improvement, as clearly indicated by the reviewers..  Please address all their comments carefully. Reviewer 1 has raised several important methodologic issues, including the potential use of interrupted time series analysis. Reviewer 2 has identified key issues with respect to interpretation of your findings. In particular, and as they pointed out, reduced admissions for NCDs reflect not only social distancing measures, and reluctance to seek care; they also can reflect diminished availability of hospital beds for admission of patients for NCDs without COVID, if most beds are devoted to COVID patients. Their point about NCD patients being admitted with COVID is also important--a person who requires hospitalization for congestive heart failure and has COVID may "officially" be counted as a COVID admission, but they have congestive heart failure (potentially worsened/precipitated by COVID). It would also be essential to articulate clearly what your analysis adds to existing knowledge. 

We look forward to receiving your revised manuscript.

Kind regards,

Kevin Schwartzman

Academic Editor

PLOS ONE

Journal Requirements:

2. We note that Figure 1 in your submission contain map image which may be copyrighted. All PLOS content is published under the Creative Commons Attribution License (CC BY 4.0), which means that the manuscript, images, and Supporting Information files will be freely available online, and any third party is permitted to access, download, copy, distribute, and use these materials in any way, even commercially, with proper attribution. For these reasons, we cannot publish previously copyrighted maps or satellite images created using proprietary data, such as Google software (Google Maps, Street View, and Earth). For more information, see our copyright guidelines: http://journals.plos.org/plosone/s/licenses-and-copyright.

Reviewers' comments:

Reviewer's Responses to Questions

**Comments to the Author**

1. Is the manuscript technically sound, and do the data support the conclusions?

Reviewer #1: Yes

Reviewer #2: Partly

2. Has the statistical analysis been performed appropriately and rigorously? 

Reviewer #1: Yes

Reviewer #2: Yes

3. Have the authors made all data underlying the findings in their manuscript fully available?

Reviewer #1: Yes

Reviewer #2: No

4. Is the manuscript presented in an intelligible fashion and written in standard English?

Reviewer #1: Yes

Reviewer #2: Yes

5. Review Comments to the Author

Reviewer #1: The reviewer thanks the authors for submission of this manuscript, entitled “Analysis of the impact of coronavirus disease 19 on hospitalization rates for chronic non-communicable diseases in Brazil”, which compares hospitalization rates pre- and during-pandemic via Poisson regression.

Below are a few points to consider to improve the manuscript:

Abstract:

• It would be helpful for the reader if the pre- vs during-pandemic period (as relevant to the Brazilian setting) were defined in the abstract

• Lines 47-48, that these declines in hospitalizations were due to “social distancing measures” is somewhat vague - might be clearer to say due to increased barriers to care during the pandemic, transport restrictions during lockdowns, etc.

• Include confidence intervals for estimates in abstract.

Background:

• Line 51: “third leading country” - this is a bit unclear - 3rd highest number of cases? Deaths? Per capita cases / deaths?

Methods:

• Line 168: definition of start of pandemic period: In the background, it was stated that restrictions were imposed following the declaration of the pandemic on 11 March 2020 - if this is the case, it’s unclear to me why the pandemic period in the analysis is then defined as starting on 1 March 2020. Were significant restrictions imposed in some areas in Brazil prior to March 11? If so, this should be clarified in the background section. If not, then it might be better to define the pandemic period as starting on March 11 rather than March 1, to align with when restrictions were actually imposed and thus began to represent barriers to care. (Otherwise, if defining the start of the pandemic period prior to imposition of actual restrictions, you may bias your estimate towards the null).

• Consider alternative methods that may be more suitable for answering this question - when analysing the impact of large-scale population-level events (i.e. a pandemic, a country-wide health policy change, etc), interrupted time series (ITS) analysis can be a useful method, as it adjusts for secular trends (other trends in the outcome that are unrelated to the event/intervention/pandemic), by fitting 2 separate regression lines to capture the deviation of the post-intervention data from its pre-intervention trend. This would allow you to isolate the effect of the pandemic on hospitalization rates from other health systems level factors that may have changed over time and affected hospitalization rates. ITS is not always an appropriate method as it requires large sample sizes over multiple time points, however, you seem to have an adequate amount of data and number of time points to make this feasible, so it is worth exploring. Otherwise, you could briefly state why ITS has not been used, if you feel it is not appropriate.

• Accounting for missing data: the issue of missing data only comes up in the discussion, where a high proportion of missing data is mentioned. It should be made clearer what proportion of data are missing, and on which variables, and methods to account for missing data should be considered (see comments under “discussion”).

• Your current analysis is not really a time series, as you are not looking at the outcome over a series of different time points, but rather, you are grouping time into two categories (pre- vs during pandemic). So, if you choose to keep your initial analysis, I would recommend not referring to it as a time series.

Results:

• Table 1: Include confidence intervals for % change (also applies to subsequent tables)

Discussion:

• Could add brief comment on assumptions of Poisson and that they have been met (otherwise alternatives e.g. Negative binomial may be more suitable)

• Line 449: in the section discussing that private sector hospitalizations were not included, it would be good to add more detail on in what ways private vs. public sector patients may differ from one another (if known) (e.g. differences in socioeconomic status?), to get more insight into how this might bias the estimate

• Missing data: a “high incidence” of missing data for “many variables” is mentioned - this is unclear - how much missing data (%)? And on how many variables? Depending on the amount and pattern of missingness, different methods can be applied to account for this - e.g. multiple imputation, inverse probability weighting, etc.

Reviewer #2: Thank you for submitting your manuscript, the paper is very well written and structured.

You conducted an ecological time series study having had access to Brazilian national health service records for admissions prior and during the pandemic. You utilized ICD-10 coding and grouped admissions by NCD.

In summary you report that admissions for NCDs across Brazil fell during the surge of COVID-19 admissions and this was due to implementation of social distancing measures.

The phenomenon of reduced NCD admissions, myocardial infarctions during the COVID-19 pandemic is well reported in the literature. Your study while giving large numbers and a nationwide view however lacks granularity beyond the statement of number of admissions. In my opinion several important points are missing like the surge in COVID-19 admissions involved patients also with comorbidities of the NCD groups and this represents a confounder to your data and analysis. Also importantly the pressure on the healthcare system meant there were no more beds for non COVID cases and if there were the NCD patients would be at higher risk of getting the infection and dying (saying patients with NCDs had unequal access is debatable if they are more vulnerable?). Healthcare workers were also affected by COVID reducing healthcare capacity further. I think there are more sides to the story that social distance measures were the cause of reduction in NCD admissions, it is not only the fear but also the lockdown how can patients travel to hospital if there is no transport etc.?

The limitations you give to your study need a little more work as they are incomplete, while you discuss the lack of private hospital data there are several omissions in your data i.e. knowing the proportion of NCD patients prior and during (through comorbidities) COVID-19. NCDs are not only managed by admissions to hospital but also by primary care.

Overall the study appears to be too narrow and the conclusions not necessarily substantiated, when it is likely there is a bigger picture, consider revising the discussion and broadening.

6. PLOS authors have the option to publish the peer review history of their article (what does this mean?). If published, this will include your full peer review and any attached files.

Reviewer #1: **Yes: **Lena Faust

Reviewer #2: No

---

## [Author Response · Author response to Decision Letter 0]

7 Feb 2022

Goiânia, January 18, 2022.

Dear Editor

Thank you for all suggestions from your reviewers about our manuscript “Analysis of the impact of coronavirus disease 19 on hospitalization rates for chronic non-communicable diseases in Brazil”. We agreed with the reviewer’s comments and we carefully attempted to evaluate all points, some of them had modifications in the text, while others we discussed.

Editor-in-Chief:

1. Reviewer 1 has raised several important methodologic issues, including the potential use of interrupted time series analysis.

Response: Thank you for the careful evaluation that made it possible to improve the quality of our manuscript. We believe that all suggestions were accepted or duly answered and justified.

2. Reviewer 2 has identified key issues with respect to interpretation of your findings. In particular, and as they pointed out, reduced admissions for NCDs reflect not only social distancing measures, and reluctance to seek care; they also can reflect diminished availability of hospital beds for admission of patients for NCDs without COVID, if most beds are devoted to COVID patients. Their point about NCD patients being admitted with COVID is also important--a person who requires hospitalization for congestive heart failure and has COVID may "officially" be counted as a COVID admission, but they have congestive heart failure (potentially worsened/precipitated by COVID). It would also be essential to articulate clearly what your analysis adds to existing knowledge.

Response: Thank you for the careful evaluation that made it possible to improve the quality of our manuscript. We believe that all suggestions were accepted or duly answered and justified. The discussion was reformulated in three topics and the conclusion was also reformulated, adding other causes for the reduction of hospitalizations.

Reviewer #1:

Comment: The reviewer thanks the authors for submission of this manuscript, entitled “Analysis of the impact of coronavirus disease 19 on hospitalization rates for chronic non-communicable diseases in Brazil”, which compares hospitalization rates pre- and during-pandemic via Poisson regression.

Response: Thank you for the careful evaluation that made it possible to improve the quality of our manuscript.

1. Abstract:

Comment: It would be helpful for the reader if the pre- vs during-pandemic period (as relevant to the Brazilian setting) were defined in the abstract.

Response: Thanks for your comment and suggestion. We have added the pre-pandemic and post-pandemic review period to the summary as presented below.

“In this study, the pre-pandemic period was set from January 1, 2017 to February 29, 2020 and the during-pandemic from March 1, 2020 to May 31, 2021.”

Comment: Lines 47-48, that these declines in hospitalizations were due to “social distancing measures” is somewhat vague - might be clearer to say due to increased barriers to care during the pandemic, transport restrictions during lockdowns, etc.

Response: As suggested by reviewer 2, there are many causes besides social distancing that may have caused the reduction and not just the lockdws. Therefore, we ask for permission to keep the new abstract conclusion in the abstract.

Conclusions: there was a decrease in the hospitalization rate for NCDs in Brazil during the COVID-19 pandemic in a scenario of social distancing measures and overload of health services.”

Comment: Include confidence intervals for estimates in abstract.

Response: Thanks for your comment and suggestion. We add 95.0% confidence interval in abstract, as presented below.

“There was a 27.0% (95.0%CI: -29.0; -25.0%) decrease in hospital admissions for NCDs after the onset of the pandemic compared to that during the pre-pandemic period. Decreases were found for all types of NCDs—cancer (-23.0%; 95.0%CI: -26.0; -21.0%), diabetes mellitus (-24.0%; 95.0%CI: -25.0%; -22.0%), cardiovascular diseases (-30.0%; 95.0%CI: -31.0%; -28.0%), and chronic respiratory diseases (-29.0%; 95.0%CI: -30.0%; -27.0%). 

2. Background:

• Line 51: “third leading country” - this is a bit unclear - 3rd highest number of cases? Deaths? Per capita cases / deaths?

Response: Thanks for your comment. The number of cases. We corrected it.

3. Methods:

• Line 168: definition of start of pandemic period: In the background, it was stated that restrictions were imposed following the declaration of the pandemic on 11 March 2020 - if this is the case, it’s unclear to me why the pandemic period in the analysis is then defined as starting on 1 March 2020. Were significant restrictions imposed in some areas in Brazil prior to March 11? If so, this should be clarified in the background section. If not, then it might be better to define the pandemic period as starting on March 11 rather than March 1, to align with when restrictions were actually imposed and thus began to represent barriers to care. (Otherwise, if defining the start of the pandemic period prior to imposition of actual restrictions, you may bias your estimate towards the null).

Response: thank for your comment. Yes, any regions implemented restrictions before March 11. We add in the text:

In Brazil, a national lockdown was not instituted by the federal government. However, measures to restrict the movement of people and suspend non-essential activities were gradually and distinctly enacted by governors and mayors of different Brazilian states and the Federal District (5,8). The Federal District was the first region in the country to promote social distancing immediately after the onset of the COVID-19 pandemic was declared on March 11, 2020 by suspending massive events and educational activities (5,8). Some municipalities implemented the blocking measures before the pandemic was declared in early March to contain the increase in cases (5,8). Subsequently, all Brazilian states implemented social distancing measures until the end of March 2020, including suspension of in-person events and classes, quarantine for the groups most vulnerable to serious COVID-19 outcomes, a full or partial economic standstill, transportation restrictions, or quarantine of the population (5,8). Most Brazilian states adopted these strategies for a period of 1–10 days after notification of the first COVID-19 case. Some areas, mainly states in the Northern and Northeastern regions, adopted these measures early on, even before notification of the first local case. Seven states suspended all economic activities during the first 13 days after notification of the first case in these areas (5). 

Comment: Consider alternative methods that may be more suitable for answering this question - when analysing the impact of large-scale population-level events (i.e. a pandemic, a country-wide health policy change, etc), interrupted time series (ITS) analysis can be a useful method, as it adjusts for secular trends (other trends in the outcome that are unrelated to the event/intervention/pandemic), by fitting 2 separate regression lines to capture the deviation of the post-intervention data from its pre-intervention trend. This would allow you to isolate the effect of the pandemic on hospitalization rates from other health systems level factors that may have changed over time and affected hospitalization rates. ITS is not always an appropriate method as it requires large sample sizes over multiple time points, however, you seem to have an adequate amount of data and number of time points to make this feasible, so it is worth exploring. Otherwise, you could briefly state why ITS has not been used, if you feel it is not appropriate.

Response: Thanks for your important comment. We followed an approach similar to that studied and analyzed by “Rennert-May E, Leal J, Thanh NX, Lang E, Dowling S, Manns B, et al. The impact of COVID-19 on hospital admissions and emergency department visits: A population-based study. PLoS One. 2021;16: 1–11. doi:10.1371/journal.pone.0252441” which analyzed the impact of COVID-19 on hospital admissions. The authors of the manuscript performed a negative binomial regression to analyze the impact of the pandemic. In addition, they performed a sensitivity analysis by ITS and found similar results, considering the 95%CI. The models' assumptions were tested and the following sentence was added to the text below.

“In our study, we used a Poisson regression model with robust variance as used in other studies that evaluated the impact of the COVID-19 pandemic on hospital admissions (HASAN et al., 2021; Wambua et al., 2021; Nourazari et al. , 2021; Filippo et al., 2021; Libruder et al., 2021). As mentioned, this model allowed including gender, age group, region of Brazil, the dummy variable representing the impact of the COVID-19 pandemic (“0”, pre-pandemic period; “1”, pandemic period) and the month of admission to control for possible seasonal variations as dependent variables. Studies show that the Poisson regression model is more suitable for time series count data when compared to the classical interrupted time series (IST) model (Bernal et al., 2017). In addition, comparative analysis showed similar results between this model and the STI in the analysis of interventions (Rennert-May et al., 2021). This model was also used instead of the negative binomial model since the data for all outcomes are not overdispersed. All assumptions of the Poisson model were met: (i) dependent variable - count per unit of time and/or space; (ii) observations independent of each other; (iii) the distribution of counts follows a Poisson distribution; (iii) model mean and variance are similar, without overdispersion. Overdispersion was analyzed for all dependent variables by Person Chi-Square for dispersion (model values ranged from 0.99 to 1.04, indicating the absence of overdispersion) (Gardner et al., 1995).”

Comment: Accounting for missing data: the issue of missing data only comes up in the discussion, where a high proportion of missing data is mentioned. It should be made clearer what proportion of data are missing, and on which variables, and methods to account for missing data should be considered (see comments under “discussion”).

Response: Thanks for your comment. In the present study, we did not use missing data, as the variables age, ICD-10 and gender that were important to extract the data were complete and with 100% coverage in the secondary database. What we mean is that due to the high percentage of missing data for the variables race and education (> 20.0%) we were not able to perform a sensitivity analysis as we did for regions, sex and age with complete data. We clarify this point in the discussion, as presented below.

“Third, there was a high incidence of missing data for many variables in dataset, such as race, and education level, which made it impossible to use these variables in the present study to analyze the impact according with these variables. “

Comment: Your current analysis is not really a time series, as you are not looking at the outcome over a series of different time points, but rather, you are grouping time into two categories (pre- vs during pandemic). So, if you choose to keep your initial analysis, I would recommend not referring to it as a time series.

Response: Thanks for your observation. We changed the design only to "ecological" due to the characteristic of the analyzed (aggregated) data.

4. Results:

Comment: • Table 1: Include confidence intervals for % change (also applies to subsequent tables).

Response: We add this.

5. Discussion:

Comment: Could add brief comment on assumptions of Poisson and that they have been met (otherwise alternatives e.g. Negative binomial may be more suitable).

Response: Thanks for the excellent comment. We have added this discussion in the Statistical Analysis of Material and Methods section as presented below:

“In our study, we used a Poisson regression model with robust variance as used in other studies that evaluated the impact of the COVID-19 pandemic on hospital admissions (HASAN et al., 2021; Wambua et al., 2021; Nourazari et al. , 2021; Filippo et al., 2021; Libruder et al., 2021). As mentioned, this model allowed including gender, age group, region of Brazil, the dummy variable representing the impact of the COVID-19 pandemic (“0”, pre-pandemic period; “1”, pandemic period) and the month of admission to control for possible seasonal variations as dependent variables. Studies show that the Poisson regression model is more suitable for time series count data when compared to the classical interrupted time series (IST) model (Bernal et al., 2017). In addition, comparative analysis showed similar results between this model and the STI in the analysis of interventions (Rennert-May et al., 2021). This model was also used instead of the negative binomial model since the data for all outcomes are not overdispersed. All assumptions of the Poisson model were met: (i) dependent variable - count per unit of time and/or space; (ii) observations independent of each other; (iii) the distribution of counts follows a Poisson distribution; (iii) model mean and variance are similar, without overdispersion. Overdispersion was analyzed for all dependent variables by Person Chi-Square for dispersion (model values ranged from 0.99 to 1.04, indicating the absence of overdispersion) (Gardner et al., 1995).”

Comment: Line 449: in the section discussing that private sector hospitalizations were not included, it would be good to add more detail on in what ways private vs. public sector patients may differ from one another (if known) (e.g. differences in socioeconomic status?), to get more insight into how this might bias the estimate.

Response: Thanks for your comment and suggestion. We have added/rephrased the discussion on the impact of the absence of private sector data on our results.

“Second, individuals in the private system may present different characteristics from those hospitalized in the public system (e.g., distribution by age, race and sex). The private sector in Brazil has a higher proportion of female, older, white and more educated users, therefore with a higher socioeconomic level when compared to the public sector (Unified Health System) (39). The private sector encompasses a vast diversification of hospital services and an even greater concentration of revenue compared to the public sector. Therefore, it would be impossible to generalize the results of the study for the entire Brazilian population and these differences, if analyzed, may impact the estimates.”

Comment: Missing data: a “high incidence” of missing data for “many variables” is mentioned - this is unclear - how much missing data (%)? And on how many variables? Depending on the amount and pattern of missingness, different methods can be applied to account for this - e.g. multiple imputation, inverse probability weighting, etc.

Response: Thanks for your comment. In the present study, we did not use missing data, as the variables age, ICD-10 and gender that were important to extract the data were complete and with 100% coverage in the secondary database. What we mean is that due to the high percentage of missing data for the variables race and education (> 20.0%) we were not able to perform a sensitivity analysis as we did for regions, sex and age with complete data. We clarify this point in the discussion, as presented below.

“Third, there was a high incidence of missing data for many variables in dataset, such as race, and education level, which made it impossible to use these variables in the present study to analyze the impact according with these variables. “

Reviewer #2:

Comment: You conducted an ecological time series study having had access to Brazilian national health service records for admissions prior and during the pandemic. You utilized ICD-10 coding and grouped admissions by NCD. In summary you report that admissions for NCDs across Brazil fell during the surge of COVID-19 admissions and this was due to implementation of social distancing measures.

Response: Thank you for the careful evaluation that made it possible to improve the quality of our manuscript.

Comment: The phenomenon of reduced NCD admissions, myocardial infarctions during the COVID-19 pandemic is well reported in the literature. Your study while giving large numbers and a nationwide view however lacks granularity beyond the statement of number of admissions. In my opinion several important points are missing like the surge in COVID-19 admissions involved patients also with comorbidities of the NCD groups and this represents a confounder to your data and analysis. Also importantly the pressure on the healthcare system meant there were no more beds for non COVID cases and if there were the NCD patients would be at higher risk of getting the infection and dying (saying patients with NCDs had unequal access is debatable if they are more vulnerable?). Healthcare workers were also affected by COVID reducing healthcare capacity further. I think there are more sides to the story that social distance measures were the cause of reduction in NCD admissions, it is not only the fear but also the lockdown how can patients travel to hospital if there is no transport etc.?

Response: Thanks for your comment. We rephrase the discussion. It includes, in addition to the blocking measures (which include measures to reduce mobility), other factors evidenced in the literature as presented below.

Comment: The limitations you give to your study need a little more work as they are incomplete, while you discuss the lack of private hospital data there are several omissions in your data i.e. knowing the proportion of NCD patients prior and during (through comorbidities) COVID-19. NCDs are not only managed by admissions to hospital but also by primary care.

Response: Obrigado pelo seu comentário e sugestões que contribuiram para melhoria da interpretação de nossos dados a luz das limitações. Reformulamos as limitações conforme apresentado abaixo:

“This study has some limitations. First, the study only included hospitalizations of individuals reported in the SUS and did not include hospitalizations in the private system. Although the SIH-SUS covers 70.0% of Brazilian hospitalizations (38), the non-inclusion of hospitalizations in the private system may have led to underestimation of hospitalization rates. As these data were not included in our analysis of the impact of the pandemic on the hospitalization rate for NCDs, the results of this study may be under- or overestimated. Second, individuals in the private system may present different characteristics from those hospitalized in the public system (e.g., distribution by age, race and sex). The private sector in Brazil has a higher proportion of female, older, white and more educated users, therefore with a higher socioeconomic level when compared to the public sector (Unified Health System) (39). The private sector encompasses a vast diversification of hospital services and an even greater concentration of revenue compared to the public sector. Therefore, it would be impossible to generalize the results of the study for the entire Brazilian population and these differences, if analyzed, may impact the estimates. Third, there was a high proportion of missing data for many variables in dataset, such as race, and education level, which made it impossible to use these variables in the present study to analyze the impact according with these variables. Thus, it was impossible to analyze the sensitivity of the pandemic’s impact according to these variables. Fourth, this study did not analyze the specific CID-10 in each NCD subgroup. The pandemic’s impact may differ for specific groups of diseases and chronic conditions (e.g., hypertension, specific cancers, and COPD). Five, important confounders were not included in the analysis, such as the number of hospital beds, number of health professionals, doctors and nurses per 1,000 inhabitants, degree of social distancing measures in the regions, including specific restrictions and prevalence of NCDs in the regions, for example. Sixth, we do not have data on the prevalence and characteristics of patients with NCDs before and during the COVID-19 pandemic, according to a large group of causes, which makes it difficult to analyze that the results of this study are due only to social distancing measures. Finally, this study only included hospital admissions and not admissions to other levels of care (eg, Primary Health Care).”

Comment: Overall the study appears to be too narrow and the conclusions not necessarily substantiated, when it is likely there is a bigger picture, consider revising the discussion and broadening.

Response: Thank you for the careful evaluation that made it possible to improve the quality of our manuscript. We believe that all suggestions were accepted.

---

## [Decision Letter · Decision Letter 1]

2 Mar 2022

Analysis of the impact of coronavirus disease 19 on hospitalization rates for chronic non-communicable diseases in Brazil

PONE-D-21-38179R1

Dear Dr. Guimarães,

We’re pleased to inform you that your manuscript has been judged scientifically suitable for publication and will be formally accepted for publication once it meets all outstanding technical requirements.

Kind regards,

Kevin Schwartzman

Academic Editor

PLOS ONE

Additional Editor Comments (optional):

None

Reviewers' comments:

Reviewer's Responses to Questions

**Comments to the Author**

1. If the authors have adequately addressed your comments raised in a previous round of review and you feel that this manuscript is now acceptable for publication, you may indicate that here to bypass the “Comments to the Author” section, enter your conflict of interest statement in the “Confidential to Editor” section, and submit your "Accept" recommendation.

Reviewer #1: All comments have been addressed

Reviewer #2: All comments have been addressed

2. Is the manuscript technically sound, and do the data support the conclusions?

Reviewer #1: Yes

Reviewer #2: Yes

3. Has the statistical analysis been performed appropriately and rigorously? 

Reviewer #1: Yes

Reviewer #2: Yes

4. Have the authors made all data underlying the findings in their manuscript fully available?

Reviewer #1: Yes

Reviewer #2: Yes

5. Is the manuscript presented in an intelligible fashion and written in standard English?

Reviewer #1: Yes

Reviewer #2: Yes

6. Review Comments to the Author

Reviewer #1: The reviewer thanks the authors for making the effort to address comments and explain/justify their methodological choices. The reviewer feels comments were adequately addressed.

Reviewer #2: Thank you for revising your manuscript, reviewing the limitations and broadening the discussion. I find the paper has improved with a more balanced discussion.

7. PLOS authors have the option to publish the peer review history of their article (what does this mean?). If published, this will include your full peer review and any attached files.

Reviewer #1: No

Reviewer #2: No

---

## [Editor Report · Acceptance letter]

14 Mar 2022

PONE-D-21-38179R1 

Analysis of the impact of coronavirus disease 19 on hospitalization rates for chronic non-communicable diseases in Brazil 

Dear Dr. Guimarães:

I'm pleased to inform you that your manuscript has been deemed suitable for publication in PLOS ONE. Congratulations! Your manuscript is now with our production department. 

Kind regards, 

on behalf of

Dr. Kevin Schwartzman 

Academic Editor

PLOS ONE